# Iron (Fe) speciation in size-fractionated aerosol particles in the Pacific Ocean: The role of organic complexation of Fe with humic-like substances in controlling Fe solubility

Kohei Sakata[1], Minako Kurisu[2], Yasuo Takeichi[3], Aya Sakaguchi[4], Hiroshi Tanimoto[1], Yusuke Tamenori[5], Atsushi Matsuki[6], Yoshio Takahashi[3,7]

[1]Center for Global Environmental Research, National Institute for Environmental Studies, 16-2 Onogawa, Tsukuba, Ibaraki 305-8506, Japan.
[2]Research Institute for Global Change, Japan Agency for Marine-Earth Science and Technology, 2-15, Natsushima-cho, Yokosuka, Kanagawa 237-0061, Japan.
[3]Institute of Materials Structure Science, High-Energy Accelerator Research Organization, Tsukuba, Ibaraki 305-0801, Japan.
[4]Faculty of Pure and Applied Science, University of Tsukuba, 1-1-1 Tennodai, Tsukuba, Ibaraki 305-8577, Japan.
[5]Japan Synchrotron Radiation Research Institute/SPring-8, 1-1-1 Kouto, Sayo, Hyogo 679-5198, Japan.
[6]Institute of Nature and Environmental Technology, Kanazawa University, Kakuma, Knazawa, Ishikawa 920-1192, Japan.
[7]Graduate School of Science, The University of Tokyo, 7-3-1 Hongo, Bunkyo-ku, Tokyo 113-0033, Japan.

*Correspondence to*: Kohei Sakata (sakata.kohei@nies.go.jp)

**Abstract**

Atmospheric deposition is one of the main sources of dissolved iron (Fe) in the ocean surfaces. Atmospheric processes are recognized as controlling fractional Fe solubility ($Fe_{sol}\%$) in marine aerosol particles. However, the impact of these processes on $Fe_{sol}\%$ remains unclear. One of the reasons for this is the lack of field observations focusing on the relationship between $Fe_{sol}\%$ and Fe species in marine aerosol particles. In particular, the effects of organic ligands on $Fe_{sol}\%$ have not been thoroughly investigated in observational studies. In this study, Fe species in size-fractionated aerosol particles in the Pacific Ocean were determined using X-ray absorption fine structure (XAFS) spectroscopy. The internal mixing states of Fe and organic carbon were investigated using scanning transmission X-ray microscopy (STXM). The effects of atmospheric processes on $Fe_{sol}\%$ in marine aerosol particles were investigated based on the speciation results. Iron in size-fractionated aerosol particles was mainly derived from mineral dust, regardless of aerosol diameter, because the enrichment factor of Fe was almost 1 in both coarse ($PM_{>1.3}$) and fine aerosol particles ($PM_{1.3}$). Approximately 80 % of the total Fe (insoluble + labile Fe) was present in $PM_{>1.3}$, whereas labile Fe was mainly present in $PM_{1.3}$. The $Fe_{sol}\%$ in $PM_{>1.3}$ was not significantly increased ($2.56 \pm 2.53$ %, $0.00–8.50$ %, n =20) by the atmospheric processes because mineral dust was not acidified beyond the buffer capacity of calcite. In contrast, mineral dust in $PM_{1.3}$ was acidified beyond the buffer capacity of calcite. As a result, $Fe_{sol}\%$ in $PM_{1.3}$ ($0.202–64.7$ %, n=10) was an order of magnitude higher than that in $PM_{>1.3}$. The $PM_{1.3}$ contained ferric organic complexes with humic-like substances (Fe(III)-HULIS, but not Fe-oxalate complexes), whose abundance correlated with $Fe_{sol}\%$. Iron(III)-HULIS was formed during transport in the Pacific Ocean because Fe(III)-HULIS was not found in aerosol particles in Beijing and Japan. The pH estimations of mineral dust in $PM_{1.3}$ established that Fe was solubilized by proton-promoted dissolution under highly acidic conditions (pH < 3.0), whereas Fe(III)-HULIS was stabilized under moderately acidic conditions (pH 3.0–6.0). Since the observed labile Fe concentration could not be reproduced by proton-promoted dissolution under moderately acidic conditions, the pH of mineral dust increased after proton-promoted dissolution. The cloud process in the marine atmosphere increases the mineral dust pH because the dust particles are covered with organic carbon and Na. The precipitation of ferrihydrite was suppressed by Fe(III)-HULIS owing to its high water solubility. Thus, the organic complexation of Fe with HULIS plays a significant role in the stabilization of Fe that was initially solubilized by proton-promoted dissolution.

## 1. Introduction

Primary production on the ocean surface is limited by the depletion of dissolved iron (Fe, Martin and Fitzwater, 1988; Jickells et al., 2005; Baker et al., 2016, 2021; Mahowald et al., 2018; Meskhidze et al., 2019). The fertilization of Fe in the surface ocean has the potential to regulate global climate systems through the uptake of atmospheric carbon dioxide ($CO_2$) in surface seawater. Dissolved Fe must be supplied to activate biological activity because microorganisms utilize dissolved Fe as a micronutrient (Boyd et al., 2007; Moore et al., 2013; Mahowald et al., 2018). Atmospheric deposition of Fe in mineral dust is a dominant source of dissolved Fe on the ocean surface (Jickells et al., 2005; Baker et al., 2016, 2021; Mahowald et al., 2018; Meskhidze et al., 2019). However, fractional Fe solubility ($Fe_{sol}\%$ = (labile Fe/total Fe) $\times$ 100) in mineral dust in source regions is usually below 1.0 % because Fe in mineral dust is typically present as insoluble species (e.g., Fe in aluminosilicates and Fe (hydr)oxides). In contrast, a wide range of $Fe_{sol}\%$ in marine aerosol particles (0.1–90 %) has been reported in previous observational studies (Buck et al., 2006; 2010; 2013, Baker and Jickells, 2006; Bakers et al., 2016, 2021; Chance et al., 2015; Kurisu et al., 2021). One of the reasons for the high $Fe_{sol}\%$ in marine aerosol particles is pyrogenic Fe with high $Fe_{sol}\%$ (up to 80 %, Schroth et al., 2009; Takahashi et al., 2013; Kurisu et al., 2016; 2019, 2021; Conway et al., 2019). It seems that the variation in $Fe_{sol}\%$ in marine aerosol particles can be explained by a binary mixing system of mineral dust and anthropogenic aerosols if the $Fe_{sol}\%$ of these components at the time of emission is known. However, explaining the variation of $Fe_{sol}\%$ in marine aerosol particles by the mixing system is difficult because atmospheric processes during transport affect the $Fe_{sol}\%$ of mineral dust and anthropogenic Fe.

The atmospheric processes of Fe are described as proton-promoted, ligand-promoted, and photo reductive Fe dissolutions (Bakers et al., 2016, 2021; Mahowald et al., 2018; Meskhidze et al., 2019 and references therein). Proton-promoted Fe dissolution is driven mainly by aerosol acidification (Desboufs et al., 1999; Mackie et al., 2005; Cwiertny et al., 2008; Shi et al., 2009, 2011, 2015; Maters et al., 2016). As a proof of the acidification of Fe-bearing particles, single-particle analysis revealed that internal mixing of Fe with sulfate, nitrate, and chloride was identified in the atmosphere (Sullivan et al., 2007; Moffet et al., 2012; Fitzgerald et al., 2015; Li et al., 2017), but these analytical techniques could not establish a direct relationship between the internal mixing state, aerosol pH and $Fe_{sol}\%$. Therefore, aerosol pH is usually estimated using thermodynamic model calculations (e.g., E-AIM and ISOROPPIA). The dissolution of Fe from aerosol particles is enhanced in the wet aerosol phase under highly acidic conditions (pH < 3.0, Longo et al., 2016; Fang et al., 2017; Tao and Murphy, 2019). However, the pH values calculated by the thermodynamic models do not necessarily reflect the pH of the mineral dust. One of the reasons for this is that the calculated result is the pH of the main component of marine aerosols (e.g., sulfate aerosols and sea spray aerosols), which are usually externally mixed with Fe-bearing particles. Another reason is that the aerosol pH of proton-promoted dissolution cannot uniquely determine the aerosol pH because the Fe-bearing particles may undergo pH cycles according to evaporation–condensation cycles. Therefore, evaluating the average pH of Fe-bearing particles for proton-promoted dissolution based on the $Fe_{sol}\%$ and/or labile (L-Fe) concentrations is appropriate.

In ligand-promoted and photo reductive Fe dissolutions, organic ligands play a significant role in enhancing $Fe_{sol}\%$ in marine aerosol particles. The formation of organic complexes on the surface of Fe oxides destabilizes the Fe−O bonds (Wang et al., 2017). Moreover, the formation of organic complexes with L-Fe in the aqueous phase promoted further Fe dissolution from the aerosol particles to aerosol liquid water (ALW). The photoreduction of Fe(III)-organic complexes also decreases the saturation index of Fe(III) in ALW because of the formation of Fe(II) (Chen and Grassian, 2013). As a result of these interactions between Fe and organic ligands, the dissolution of Fe-bearing particles is promoted. Oxalate is considered an important ligand in aerosol particles because oxalate is ubiquitously present in aerosol particles. However, the mass fraction of oxalate in water-soluble organic carbon (WSOC) is typically lower than 10 % (Bikkina et al., 2015; Kawamura and Bikkina, 2016). In contrast, more than half of WSOC is present as humic-like substances (HULIS), which are considered to affect $Fe_{sol}\%$ in aerosol particles (Wozniak et al., 2013, 2015; Al-Abdleh 2015). Atmospheric HULIS in marine aerosols are formed by atmospheric processes and direct emissions from the ocean surface (Deng et al., 2014; Chen et al., 2016; Santander et al., 2021), whereas soil-derived organic matter is generally not an important source of atmospheric HULIS (Graber and Rudich, 2006; Spranger et al., 2020). In addition, siderophores have been detected in aerosols, rainwater, and cloud water, which are likely formed by biological activities in mineral dust and cloud water (Cheize et al., 2012; Sullivan et al., 2012; Vinatier et al., 2016). The siderophore has a higher stability constant with Fe than with oxalate, and Fe-siderophore complexes have high water solubility (Cheize et al., 2012). Recently, Fe(III)-dextran as Fe(III)-organic complexes were detected in $PM_{2.5}$ collected in Colorado, USA (Salazar et al., 2020). The formation of Fe-organic complexes may suppress the precipitation of nano-ferrihydrite when acidified aerosol particles with high $Fe_{sol}\%$ encounter high-pH solutions because these Fe-organic complexes have higher water solubility than inorganic Fe over a wide pH range. However, the effects of Fe(III)-organic complexes of HULIS and siderophores in atmospheric samples on $Fe_{sol}\%$ have not been well investigated through field observations of marine aerosol particles.

This was a case study on the relationship between $Fe_{sol}\%$ and Fe species in size-fractionated aerosol particles collected from the Pacific Ocean. The iron species in the aerosol samples were determined using X-ray absorption fine structure (XAFS) spectroscopy to investigate the relationship between Fe species and $Fe_{sol}\%$. XAFS spectroscopy provides the average fraction of Fe species, which can be directly compared to the $Fe_{sol}\%$. In addition, the Al species in several size-fractionated aerosol particles were determined for evaluating the aging effect of the aluminosilicates in the samples. The Al K-edge X-ray absorption near-edge structure (XANES) spectrum is sensitive to the coordination chemistry of Al (Ildefonse et al., 1998; Shaw et al., 2009; Hagvall et al., 2015). Furthermore, the internal mixing states of Fe and organic carbon (OCs) were investigated using scanning transmission X-ray microscopy (STXM) for evaluating the detailed alteration processes of Fe-bearing particles. Based on the $Fe_{sol}\%$ and speciation results, the expected pH required for L-Fe concentration in the aerosol samples by proton-promoted dissolution within the transport time ($pH_{PPD}$) was evaluated using a conceptual model following first-order iron dissolution. In addition to $pH_{PPD}$, pH for stabilization of L-Fe species in aerosol particles ($pH_{L-Fe}$) was evaluated by a geochemical model. If $pH_{L-Fe}$ differs from $pH_{PPD}$, L-Fe species are formed under different pH conditions from proton-promoted dissolution. Therefore, the differences between $pH_{PPD}$ and $pH_{L-Fe}$ may be an indicator of the pH

variation of the Fe-bearing particles. From these results, the role of atmospheric processes for enhancing $Fe_{sol}$% was discussed in this study.

## 2. Sampling and analytical methods

### 2.1. Aerosol sampling

Size-fractionated sampling of marine aerosols was conducted during the research cruise of *R/V Hakuho-Maru* (Fig. 1 and Table S1: KH-14-6, longitudinal cruise of the Pacific Ocean, December 2, 2014, to February 26, 2015, GEOTRACES).
Three size-fractionated aerosol particles were collected from the western Pacific Ocean (WPO), and one sample was collected from the central Pacific Ocean (CPO) and southern Pacific Ocean (SPO, Fig. 1). A high-volume air sampler (MODEL-123SL, Kimoto, Japan) with a Sierra-type cascade impactor (TE-236, Tisch Environmental Inc., USA) was installed on the compass deck of the vessel located 13 m above sea level. The sampling airflow rate was set at 0.566 $m^3$/min. The wind speed and direction were monitored using a wind-sector control system to prevent the contamination of fly ash and
exhaust gases emitted from the vessel. Aerosol samples were stored in a dry desiccator at 20 % relative humidity and room temperature (approximately 20 ºC). Aerosol particles were collected in seven stages, of which aerodynamic diameters were >10.2 μm (stage-1: S1), 4.2–10.2 μm (stage-2: S2), 2.1–4.2 μm (stage-3: S3), 1.3–2.1 μm (stage-4: S4), 0.69–1.3 μm (stage-5: S5), 0.39–0.69 μm (stage-6: S6), and <0.39 μm (stage-7: S7). Aerosol samples in S1 to S4 were defined as coarse aerosol particles ($PM_{>1.3}$), whereas S5 to S7 were defined as fine aerosol particles ($PM_{1.3}$). Aerosol particles in S1–S6 were collected
on a custom-built polytetrafluoroethylene (PTFE, approximately 15 $cm^2$) filter (Sakata et al. 2018). The PTFE filter was rinsed using the following procedures with heating at 150 °C: ultrapure water (MQ, Merck Millipore, USA), 3 mol/L $HNO_3$ (Electric grade, Kanto Chemical, Japan), 3 mol/L HCl (Electric grade, Kanto Chemical, Japan), and MQ water (Sakata et al., 2018). The Al and Fe blanks in the PTFE filter were 0.306 ± 0.352 and 0.335 ± 0.340 ng/$cm^2$, respectively. The unit of the filter blank concentration was converted from ng/$cm^2$ to ng/$m^3$ using the following equation:

$$Filter\ blank\ (ng/m^3) = \frac{filter\ blank\ (ng/cm^2) \times filter\ area\ (cm^2)}{Total\ flow\ for\ each\ sampling\ (m^3)}\ (Eq.\ 1)$$

As a result, the blank concentrations of Al and Fe were a few pg/$m^3$. The blank concentrations of Fe and Al were approximately one order of magnitude lower than the lowest concentrations of these elements in the samples. For single-particle analysis, aerosol particles were collected on molybdenum grids with a formvar thin film (Mo grid) fixed on the PTFE filter using double-face cellulose tape. Aerosol samples from S7 were collected on a cellulose filter (Whatman 41, 516
$cm^2$, GE Healthcare, USA). The filter blank of Al and Fe in the cellulose filter was 7.20 and 16.5 ng/$cm^2$, which corresponded to 2.52 and 5.77 ng/$m^3$, respectively. Stage 7 was excluded from the discussion because of its high-filter background. In this study, the sample names are described as the stage number of the cascade impactor combined with the sampling site (e.g., stage 6 collected in SPO: S6-SPO).

Aerosol sampling was performed at the Noto Ground-based Research Observatory (NOTOGRO) located in the coastal region of the Sea of Japan (Suzu, Ishikawa, Japan: 37.4513 °N, 137.3589 °E). NOTOGRO is located between China and the sampling sites in the WPO (Fig. 1). Size-fractionated aerosol samples influenced by Chinese air masses were collected from February 19 to 26, 2020 (Fig. S1a). In addition, the reference material of Beijing aerosol (NIES CRM 28, Urban dust, Mori et al., 2008) was also employed for comparing Fe species.

## 2.2. Total and labile metal concentrations

All sample treatments were conducted in a clean booth (Class-100) and evaporation chamber installed in a Class-10000 clean room. Acid digestion and ultrapure water extraction of aerosol samples were performed for determining total and labile metal concentrations, respectively. Aerosol samples were decomposed using mixed acid (2 mL of 15.2 mol/L $HNO_3$, 2 mL of 9.3 mol/L HCl, and 1 mL of 20 mol/L HF) and heated at 120 °C for 1 d. The mixed acid was evaporated to dryness at 120 °C, and the residues were re-dissolved in 0.15 mol/L $HNO_3$. Labile metals in the aerosol particles were extracted ultrasonically for 30 min using 5 mL of MQ water. The extracted solutions were acidified to 0.15 mol/L after filtration of insoluble particles using a hydrophilic syringe PTFE filter ($\phi$:0.20 μm, Dismic®, 25HP020AN, Advantec, Japan). Total and labile metal concentrations were determined using inductively coupled plasma mass spectrometry (ICP-MS, Agilent 7700, Agilent, Japan). Total and labile metal concentrations in total suspended particulates (TSP) were calculated by summing target metal concentrations in stages 1 to 6. The fractional Fe and Al solubility ($Fe_{sol}$% and $Al_{sol}$%, respectively) and enrichment factors (EF) were calculated using the following equations:

$$Fe_{sol}\% = (labile\ Fe/total\ Fe) \times 100, (Eq.\ 2)$$
$$Al_{sol}\% = (labile\ Al/total\ Al) \times 100, (Eq.\ 3)$$
$$EF = (Fe/Al)_{aerosol}/(Fe/Al)_{crust}. (Eq.\ 4)$$

The Fe and Al concentrations in the continental crust were referred from Taylor (1964).

## 2.3. Major ion and WSOC concentrations

The major ions ($Na^+$, $NH_4^+$, $K^+$, $Mg^{2+}$, $Ca^{2+}$, $Cl^-$, $NO_3^-$, $SO_4^{2-}$, and $C_2O_4^{2-}$) in the aerosol samples were extracted using the same methods for labile metal extraction. The major ion concentrations were measured using ion chromatography (ICS-1100, Dionex, Japan). The guard and separation columns for cations were Ion Pac CG12A and CS12A, respectively, and those for anions were Ion Pac AG22 and AS22, respectively. The guard and separation columns were installed in a thermo-controlled box (30 °C). The eluents for cations and anions were 20 mmol/L of methanesulfonic acid and a mixed solution of 4.5 mmol/L $Na_2CO_3$/1.4 mmol/L $NaHCO_3$. After passing through the column, the eluents were passed through a suppressor and were introduced into the conductivity detector. The detection limits of the ICS-1000 for $Na^+$, $NH_4^+$, $K^+$, $Mg^{2+}$, $Ca^{2+}$, $Cl^-$, $NO_3^-$, $SO_4^{2-}$, and $C_2O_4^{2-}$ were 0.556, 0.464, 1.15, 0.726, 1.50, 5.62, 15.0, 18.8, and 33.2 ng/mL, respectively. Among the targeted ions, the lowest and highest filter blank concentrations were 0.0687 and 32.4 $ng/cm^2$ for $Mg^{2+}$ and $SO_4^{2-}$,

respectively (Sakata et al., 2018). After the unit conversion of the filter blank from $ng/cm^2$ to $ng/m^3$ using Equation 1, the highest filter blank concentration was 4.47 $ng/m^3$ $SO_4^{2-}$. Semi-volatile compounds (e.g., $NH_4NO_3$) were affected by negative artifacts during sampling. The negative artifact effect was unlikely to be significant because most nitrates were present in $PM_{>1.3}$ with a small concentration of $NH_4^+$. However, some $NH_4NO_3$ present in $PM_{1.3}$ may be affected by the negative artifact. The negative artifacts of oxalate and ammonium sulfate are usually negligible in IC analyses (Yao et al., 2002; Bian et al., 2014). The non-sea salt (nss) $SO_4^{2-}$ and $Ca^{2+}$ were calculated using the following equation:

$$[nss\text{-}SO_4^{2-} \text{ or } nss\text{-}Ca^{2+}] = [SO_4^{2-} \text{ or } Ca^{2+}]_{aerosol} - [Na^+]_{aerosol} \times ([SO_4^{2-} \text{ or } Ca^{2+}]/[Na^+])_{seawater} \text{ (Eq. 5)}$$

WSOC was extracted using 15 mL of MQ water in glass vials on a shaker for 1 h, and then the WSOC concentrations were measured using a total carbon analyzer (TOC-V CSH, Shimadzu, Japan).

## 2.4. Estimation of available proton for mineral dust ($[H^+]_{mineral}$)

The available protons for mineral dust ($[H^+]_{mineral}$) were estimated using the following procedures for evaluating the degree of acidification of mineral dust in aerosol particles. First, $NO_3^-$ and $nss\text{-}SO_4^{2-}$ concentrations other than ammonium salts ($[NO_3^- \text{ and } nss\text{-}SO_4^{2-}]_{non\text{-}NH4}$) were estimated using the following equation, assuming that $[NH_4^+]_{neq}$ was present as $NH_4NO_3$ and $(NH_4)_2SO_4$:

$$[NO_3^- \text{ and } nss\text{-}SO_4^{2-}]_{non\text{-}NH4} = [NO_3^-] + 2\times[nss\text{-}SO_4^{2-}] - [NH_4^+] \text{ (Eq. 6)}$$

Subsequently, $NO_3^-$ and $nss\text{-}SO_4^{2-}$ associated with $Na^+$ in sea spray aerosols (SSA) were estimated. $NaNO_3$ and $Na_2SO_4$ are formed by chlorine depletion of SSA, as described in the following chemical reactions:

$$NaCl (aq) + HNO_3 (g) \rightarrow NaNO_3 (aq) + HCl (g), \text{ (R1)}$$

$$2NaCl (aq) + H_2SO_4 (aq) \rightarrow Na_2SO_4 (aq) + 2HCl (g), \text{ (R2)}$$

Thus, molar concentrations of $NaNO_3 + Na_2SO_4$ were equivalent to chlorine depletion from the SSA. Therefore, $[NO_3^-]_{neq}$ and $[nss\text{-}SO_4^{2-}]_{neq}$ combined with $Na^+$ were estimated using the following equations:

$$[Cl^- \text{ loss}] = [NaNO_3] + [Na_2SO_4] = ([Cl^-]_{seawater}/[Na^+]_{seawater}) \times [Na^+]_{aerosol} - [Cl^-]_{aerosol} \text{ (Eq. 7)}$$

Assuming that $NO_3^-$ and $nss\text{-}SO_4^{2-}$ other than ammonium and Na salts were derived from the heterogeneous reactions of $HNO_3$ and $H_2SO_4$ with mineral dust (e.g., $CaCO_3$), we evaluated the available acids for mineral dust ($[H^+]_{mineral}$) using the following equation:

$$[H^+]_{mineral} = [NO_3^- \text{ and } nss\text{-}SO_4^{2-}]_{mineral} = [NO_3^- \text{ and } nss\text{-}SO_4^{2-}]_{non\text{-}NH4} - [Cl^- \text{ loss}], \text{ (Eq. 8)}$$

Here, $[H^+]_{mineral}$ refers to the maximum amount of strong acids ($H_2SO_4$ and $HNO_3$) that can be internally mixed with Fe-bearing particles and does not guarantee that all $[H^+]_{minerals}$ are internally mixed with the mineral particles. When $[H^+]_{mineral}$ is negative, the mineral dust in the aerosol sample was not well acidified. In contrast, if $[H^+]_{mineral}$ is higher than [nss-Ca] ($[H^+]_{mineral} > 0$), mineral dust has the potential to be acidified beyond the buffering capacity of $CaCO_3$.

## 2.5. Iron speciation by XAFS

The average Fe species in the aerosol samples was determined using bulk XAFS spectroscopy at BL-9A and BL-12C at the Photon Factory (PF), Ibaraki, Japan (Nomura and Koyama, 2001). The synchrotron radiation generated by a bending magnet was monochromatized using a double-crystal monochromator of Si(111). The energy resolution of the monochromator was approximately 0.2 eV. Energy calibration was performed with the peak top of the pre-edge peaks of hematite aligned to 7112 eV. Approximately 1/10 of the collected aerosol samples on the PTFE filters were transferred to double-face carbon tape oriented at 45° to the orthogonal direction of the incident X-ray beam. Iron K-edge XANES spectra of all the target samples were recorded in the fluorescence yield (FY) mode. The EXAFS spectra were simultaneously recorded with XANES for samples with sufficiently high Fe concentrations for obtaining the EXAFS spectra. The scanning energies of the XANES and EXAFS were 7080–7200 and 7080–7530 eV, respectively. All XANES and EXAFS spectra were recorded in the FY mode. Fluorescence X-rays from the samples were detected using a 19-element Ge solid-state detector equipped with a Soller slit. Spectrum analysis of the XANES and EXAFS spectra was performed using the REX2000 software (Rigaku, Japan). The energy regions for linear combination fitting (LCF) of the XANES and EXAFS spectra were 7100–7200 eV and 0–10 Å in k-space, respectively.

Reference materials for inorganic Fe are ferrihydrite, goethite, hematite, weathered biotite, chlorite, illite, smectite, montmorillonite, and Fe(III)sulfate. The details of these references are described by Takahashi et al. (2011). Iron(II)–oxalate, Fe(III)–oxalate, Fe(III)–stearate, Fe(III)–nitrate, Fe(III) complexed with deferoxamine (Fe(III)-DFO), and Fe(III)-humate were used as reference materials for Fe(III)-organic complexes. Among the Fe(III)-organic complexes, Fe(III)-citrate, Fe(III)-stearate, Fe(III)-humate, and Fe(III)-DFO are defined generally as Fe(III)-HULIS. The Fe K-edge XANES and EXAFS spectra of the key species are shown in Fig. 2. The XANES spectrum of Fe(III)-sulfate showed a small shoulder in the high-energy region of the peak at 7130 eV (Fig. 2a). Iron(III)–oxalate and hematite also have an intense peak at approximately 7130 eV and a small shoulder in the low-energy region of the peak (Fig. 2a). These species were distinguished from Fe(III)-HULIS because Fe(III)-HULIS has a flat peak at 7125–7135 eV (Fig. 2a). In the case of ferrihydrite and goethite, these XANES spectra have a flatter peak than hematite, but the width of the peak is narrower than that of Fe(III)-HULIS (Fig. 2a). Furthermore, the EXAFS spectrum of Fe(III)-HULIS was clearly different from that of ferrihydrite, goethite, and hematite. Fe(III)-HULIS has a single peak at 7–9 Å in the k-space, whereas Fe-(hydr)oxides have two peaks in the same region (Fig. 2b). Based on these spectral differences, the Fe species in the aerosol particles were determined using the LCF method.

## 2.6. Al and Na speciation by XANES spectroscopy

Al and Na speciation experiments were performed at BL-19B in PF and BL27SU in SPring-8, respectively. For both beamlines, the synchrotron radiation generated by an undulator was monochromatized using a valid line spacing plane grating monochromator (VLS-PGM). Aerosol samples on carbon tape were installed in a vacuum chamber because of the short attenuation length of soft X-rays (< 2000 eV) in the ambient atmosphere. The Al K-edge (1550–1600 eV) and Na K-

edge (1065–1100 eV) XANES spectra of the aerosol samples were recorded in the FY mode. Fluorescence X-rays were detected using a single-element silicon-drift detector.

## 2.7. Single-particle analysis

Single-particle analyses were conducted using STXM at BL-13A in PF (Takeichi et al. 2016). Monochromatic X-rays were focused at 30 nm × 30 nm using a Fresnel zone plate. The aerosol sample on the Mo grid was mounted on a piezo-controlled stage in a chamber purged with 0.1 atm He. Firstly, aerosol particles were imaged at the following energies: 280 eV (pre-edge), 285.0 eV (aromatic C), 287.6 eV (aliphatic C), 288.8 eV (carboxylic/hydroxamate C), carbonate (290.3 eV), and 297.2 eV (K L-edge), and 305 eV (post-edge). The Fe and Na distributions were identified by image subtraction of the post-edge from the pre-edge. The typical imaging size was $15 \times 15$ $\mu m^2$ with a spatial resolution of $0.06 \times 0.06$ $\mu m^2$. Subsequently, the XANES spectra of C, K, Fe, Na, and Al were obtained separately using the image-stack mode. The typical image size of the image stack was $3 \times 3$ $\mu m^2$. The image drift was aligned after appending the image-stack data of all elements. The XANES spectra of the regions of interest (e.g., the core and surface of the aerosol particles) were extracted using aXis2000 software.

## 2.8. Estimation of pH for proton-promoted dissolution (pH$_{PPD}$)

The average pH of the proton-promoted dissolution (pH$_{PPD}$) was estimated using three Fe-pool models. The model was constructed based on a previous study on dissolution experiments using Beijing dust (dust/liquid ratio: 60 mg/L), as reported by Shi et al. (2011). The three Fe pools (fast, intermediate, and slow) have different dissolution rate constants according to first-order kinetics (Shi et al., 2011). The observed L-Fe concentration of aerosol particles ([L-Fe($t$)]$_{aerosol}$) can be described by the following equation:

$$[\text{L-Fe}(t)]_{aerosol} \ (\mu mol/g) = [\text{L-Fe}(t)]_{fast} + [\text{L-Fe}(t)]_{intermediate} + [\text{L-Fe}(t)]_{slow} \ (\text{Eq. 9})$$

$$[\text{L-Fe}(t)]_{fast/intermediate/slow} \ (\mu mol/g) = [\text{obs-Fe}] \times [\%\text{Fe}(0)]_{fast/intermediate/slow} \times (1 - e^{-kt}) \ (\text{Eq. 10})$$

where $t$ is the dissolution time (h), [L-Fe($t$)]$_{fast/intermediate/slow}$ is the labile Fe concentration normalized by the mass concentration of mineral dust ($\mu mol/g$) at time $t$, [obs-Fe] is the observed total Fe concentration, [%Fe(0)]$_{fast/intermediate/slow}$ is the percentage of solubilized Fe in each pool to the total Fe, and $k$ is the dissolution rate constant for each pool. Mass concentration of mineral dust for normalizing L-Fe concentrations was estimated by total Al concentration divided by the percentage of Al in the continental crust (8.23 %). Mineral dust is expected to undergo several condensation-evaporation cycles during transport (Pruppacher and Jaenicke, 1995). Proton-promoted Fe dissolution occurred during the evaporation state (wet aerosol), whereas aerosol particles were taken in cloud water during the condensation phase. According to a previous study, the global average residence times for aerosol particles before uptake by clouds and within the cloud in an air parcel are up to 12 h and 3 h, respectively (Pruppacher and Jaenicke, 1995). Based on these residence times, aerosol particles are expected to exist in an evaporative state (wet aerosol) for approximately 67–80 % of their transport time. In this study, the estimation of pH was estimated assuming that aerosol particles spent the evaporated state in 75 % of transport time

(approximately 90 h for the WPO and 130 h for CPO and SPO). The %Fe(0) and k values for each pool are described as a function of pH (Table 1). Previous studies have assumed illite to be the major Fe species of Fe-containing minerals in the slow pool. However, weathered biotite was the dominant Fe-containing mineral in our samples (see Section 3.2). Because the dissolution rate of biotite is approximately an order of magnitude higher than that of illite (Bibi et al., 2011; Bray et al., 2015), the equation given in a previous study can be rewritten as:

$$\text{Previous study: } \log k_{slow} = -0.44 \, \text{pH} - 1.76. \text{ (Eq. 11)}$$

$$\text{This study: } \log k'_{slow} = -0.44 \, \text{pH} - 0.76. \text{ (Eq. 12)}$$

Finally, the dissolution curves at various pH values are described in Table 1. This curve with the pH of each sample was used to explain the observed L-Fe within the expected transport time. It should be noted that these kinetic parameters are estimated using the experimental data with a solid/liquid ratio of 60 mg/L. The actual aerosol dust/liquid ratios are predicted to be as high as 3000 g/L, which may suppress the dissolution of Fe from the aerosol particles (Shi et al., 2011). Our calculation results may have overestimated the modeled L-Fe concentration at $pH_{PPD}$ with a high dust/liquid ratio. In other words, a lower pH (or higher aerosol acidity) than the predicted $pH_{PPD}$ is required to account for the observed L-Fe concentration, while considering the suppression effect. Therefore, $pH_{PPD}$ can be recognized as the upper pH limit to explain the observed L-Fe concentration by proton-promoted dissolution.

## 2.9. Geochemical modeling of L-Fe species

ALW contents in $PM_{1.3}$, calculated using E-AIM Model IV (Clegg et al., 1992; Friese and Ebel, 2010), which can have an agreement of ALW between observational and modeled water mass has been reported in a previous study (Engelhart et al., 2011). The input parameters for E-AIM Model IV were the molar concentrations of $H^+$, $Na^+$, $NH_4^+$, $Cl^-$, $NO_3^-$, $SO_4^{2-}$, temperature, and relative humidity. The proton concentration was estimated using the following equation:

$$[H^+] = [Cl^-] + [NO_3^-] + 2 \times [SO_4^{2-}] - [Na^+] - [NH_4^+] \text{ (Eq. 13)}$$

In this study, the buffering effect of calcite in the equilibrium calculation was not considered because (i) mineral dust was likely acidified beyond the buffering capacity of calcite, and (ii) calcite in fine aerosol particles was altered to $CaSO_4 \cdot 2H_2O$ and $CaC_2O_4$ during transport from the source region of Asian dust to Japan (Takahashi et al., 2008; Miyamoto et al., 2020).

The L-Fe species in ALW were calculated using the React model in GWB software (Bethke, 1996). The input data were the molar concentrations of all major ions, oxalate ions, labile metals (Al, Mn, Fe, Ni, Cu, Zn, Cd, Ba, and Pb), WSOC, ALW, and temperature. The precipitation of metal species with a high saturation index (> 1.0) was inhibited in the simulation of the high-ionic-strength conditions of ALW (> few mol/L, Herrmann et al., 2015). L-Fe species were calculated under various mixing ratios of WSOC for evaluating the effects of the internal mixing state between L-Fe and WSOC on L-Fe species. The mixing ratio was 1.0 %, 2.5 %, 5.0 %, 10 %, 25 %, 50 % and 100 % of WSOC concentration. For all calculations, the thermodynamic data for citric acid with Fe were used as a representative organic ligand because the stability constant and acid dissociation constant of citric acid (log K:13.13 and pKa1:3.13) are similar to those of HULIS (log $K_{HULIS}$: 11.1–13.9 and $pK_a$: 3.3–4.0, Salma and Láng, 2008; Samburova et al., 2008; Abualhaija et al., 2015). The initial pH was

fixed at 7 and subsequently shifted from 7 to 0 for calculating the pH dependence of the L-Fe species in ALW. A total of 276 aqueous species were considered in the calculation, and the stability constants of the main species are shown in section of Thermodynamic Data in the Supplemental Information.

## 3. Results and Discussion

### 3.1. Total and labile Fe and Al concentrations in TSP

Backward trajectory analysis was performed using the hybrid single-particle Lagrangian integrated trajectory model (HYSPLIT, Stein et al., 2015). The WPO samples were affected by Asian continental outflows, whereas the air masses in the CPO and WPO were derived from the pelagic regions (Fig. S1). Total Fe and Al concentrations in TSP at the WPO samples (Figs. 3a and 3e, Fe: 75.6–257 ng/m$^3$, Al: 130–422 ng/m$^3$) were one to two orders of magnitude higher than those in the CPO and SPO (Figs. 3a and 3e, Fe: 0.733–4.37 ng/m$^3$, Al: 3.56–4.12 ng/m$^3$). Labile Fe and Al concentrations were also higher in the WPO samples (Figs. 3b and 3f, Fe: 2.61–19.8 ng/m$^3$, Al: 3.56–27.0 ng/m$^3$) than in the CPO and SPO (Figs. 3b and 3f, Fe: 0.0422–0.0489 ng/m$^3$, Al: 0.0383–0.0678 ng/m$^3$). Thus, the high total and labile metal concentrations were attributed to continental air masses (Figs. 3 and S1). The EF of Fe in these samples were 0.26–1.8 (0.92±0.55), indicating that Fe in these TSP samples was mainly derived from mineral dust (Fig. 3d). The Fe$_{sol}$% and Al$_{sol}$% in TSP were 5.30 ± 2.99 % (0.967–7.69 %) and 3.32 ± 2.22 % (1.08–6.40 %), respectively (Figs. 3c and 3g). These values were within the range reported in previous studies (Mahowald et al., 2018 and references therein).

### 3.2. Size distributions of Fe and Al concentrations

The total Fe and Al concentrations in PM$_{>1.3}$ were higher than those in PM$_{1.3}$ (Figs. 4a and 4d). Fe and Al in PM$_{>1.3}$ accounted for 78.5 ± 8.34 % (n= 5, 69.9–87.9 %) and 81.8 ± 8.53 % (n= 5, 72.0–88.8 %) in TSP, respectively. The EF of Fe and Ti as typical crustal elements were almost 1.0, regardless of aerosol diameter (Fig. S2). This is because mineral dust was mainly present at PM$_{>1.3}$. The labile concentrations of Fe and Al were higher in PM$_{1.3}$ to PM$_{>1.3}$ (Figs. 4b and 4e). Labile Fe and Al in PM$_{>1.3}$ accounted for 60.5 ± 34.1 % (n= 5, 10.7–87.3 %) and 45.9 ± 24.1 % (n= 5, 24.2–76.2 %) in TSP, respectively. Thus, the size distributions of the L-Fe and L-Al concentrations were evidently different from those of the total Fe and Al. The average Fe$_{sol}$% in PM$_{>1.3}$ and PM$_{1.3}$ were 2.56 ± 2.53 % (n=20, 0.00–8.50 %) and 22.3±21.7 % (n=10, 0.202–64.7 %), respectively (Fig. 4c). In the case of Al, the average Al$_{sol}$% in PM$_{>1.3}$ and PM$_{1.3}$ were 2.76±2.85% (n=20, 0.389–11.5 %) and 11.7±10.8% (n=10, 0.700–32.4 %), respectively (Fig. 4f). Thus, both Fe and Al in PM$_{>1.3}$ were more soluble than those in PM$_{>1.3}$. Therefore, even if the total Fe concentration in PM$_{1.3}$ was lower than that in PM$_{1.3}$, PM$_{1.3}$ plays a significant role in supplying Fe to the ocean surface.

The enrichment of labile Fe and Al in PM$_{1.3}$ has been reported in previous studies (Baker and Jickells, 2006; Buck et al., 2010; Chance et al., 2015; Baker et al., 2020; Kurisu et al., 2021). One of the reasons for the enrichment of labile Fe in PM$_{1.3}$ is the presence of anthropogenic Fe in PM$_{1.3}$ (Kurisu et al., 2016; 2021; Hsieh et al., 2022). It is known that anthropogenic Fe is emitted as Fe oxides with a small amount of coexisting elements, which should affect the EF of Fe. In fact, the EF of Fe in

$PM_{1.3}$ impacted by anthropogenic Fe, was higher than 2.0 (Kurisu et al., 2016, 2019; Hsieh et al., 2022; Zhang et al., 2022). However, the EF of Fe in our samples was almost 1.0 (Fig. S2), indicating that the influence of anthropogenic Fe oxides was insignificant. Oil combustion, including ship emissions, is one of the dominant sources of pyrogenic Fe in $PM_{1.3}$ as several studies have reported good correlations between high $Fe_{sol}\%$ and high EFs of V and Ni (Sedwick et al., 2007; Sholkovitz et al., 2009; Ito, 2015). However, $Fe_{sol}\%$ in $PM_{1.3}$ was not correlated with the EF of V and Ni as tracers of oil combustion (Figs.

5a and 5b), which is consistent with the results of an observational study of the Pacific Ocean (Buck et al., 2013).

Coal fly ash is a dominant source of Fe in marine aerosol particles (Schroth et al., 2009; Sedwick et al., 2007; Sholkovitz et al., 2009; Chen and Grassian., 2013; Ito, 2015; Baldo et al., 2022). $Fe_{sol}\%$ in $PM_{1.3}$ correlated with the EF of Pb and nss-$SO_4^{2-}$ (Figs. 5c and 5d), which is a good tracer for municipal solid waste incineration and coal combustion in terrestrial regions (Nriagu and Pacyna, 1988; Sakata et al., 2000, 2014). Since the EF of Fe in coal fly ash is close to 1,

distinguishing between mineral dust and coal fly ash based on the EF of Fe is difficult (EF was calculated using NIST reference materials, Table S2). In contrast, coal and coal fly ash tended to be enriched in Co (EF~10, Table S2). Assuming that the $Fe_{sol}\%$ (mean: 22.4%) in $PM_{1.3}$ can be associated with high soluble Fe in coal fly ash ($Fe_{sol}\%$: 100%) with high EF of Co (~10), the EF of Co in the $PM_{1.3}$ becomes approximately 3.0. Moreover, L-Fe was extracted with MQ water in this study (weakly acidic to neutral conditions), but Fe in the coal fly ash is hardly soluble under these conditions ($Fe_{sol}\% <$

0.2%, Desboefus et al., 2005; Oakes et al., 2012a). Furthermore, all of the Fe in coal fly ash is not dissolved in acidic solutions ($Fe_{sol}\%$: ~ 70% at pH 1.0, Chen and Grassian, 2013; Baldo et al., 2022). Therefore, if coal fly ash is the dominant L-Fe source, the EF of Co in the aerosol should be higher than 3.0. However, the EFs of Co in the $PM_{1.3}$ samples were approximately 1.0 (Fig. S2). These results indicated that Fe in size-fractionated aerosol particles were mainly derived from mineral dust rather than coal fly ash and anthropogenic Fe oxides. However, $Fe_{sol}\%$ in non-aged

mineral dust is usually less than 1.0% in weakly acidic and neutral solutions. Therefore, high $Fe_{sol}\%$ in $PM_{1.3}$ were caused by atmospheric processes of mineral dust during the transport.

The concentration of $[H^+]_{mineral}$ is higher than [nss-$Ca^{2+}$] in $PM_{1.3}$ with high $Fe_{sol}\%$ (>10 %). This result implies that mineral dust was acidified beyond the buffering capacity of calcite (Figs. 6a–6c). The dominant source of $[H^+]_{mineral}$ in the WPO samples was mainly $SO_2$ or $H_2SO_4$ because the $NO_3^-$ concentrations were lower than those of nss-$SO_4^{2-}$ (Fig. S3a and

365 3b). The sources of nss-$SO_4^{2-}$ in East Asia and its outflow were mainly derived from anthropogenic emissions owing to the lower S isotope ratio (about few ‰, Inomata et al., 2016; Chung et al., 2019) than that of biogenic S (18–20 ‰, Amrani et al., 2013). Previous studies have reported that the good relationship between $Fe_{sol}\%$ and nss-$SO_4^{2-}$ is attributed to the solubilization of Fe by coal-derived $SO_2$ (Fang et al., 2015; Wong et al., 2020), and a good correlation between $Fe_{sol}\%$ and nss-$SO_4^{2-}$ was found in this study (Fig. 5d). This result is consistent with previous studies because Asian dust, especially

aluminosilicate, in $PM_{1.3}$ is internally mixed with sulfate (Sullivan et al., 2007; Fitzgerald et al., 2015; Li et al., 2017; Sakata et al., 2021). Therefore, the acidification of mineral dust by coal-derived $SO_2$ or $H_2SO_4$ during transport in East Asia is the dominant reason for the high $Fe_{sol}\%$ in $PM_{1.3}$ collected in the WPO. It should be noted that the correlation between $Fe_{sol}\%$

and EF of Pb was caused by a high correlation between nss-$SO_4^{2-}$ and EF of Pb (Fig. 5e). Considering the causal relationship between $Fe_{sol}$% and EF of Pb, it is difficult to believe that $Fe_{sol}$% increases with increasing emissions of coal fly ash (increasing EF of Pb) because Fe in coal fly ash is insoluble unless the fly ash undergoes acidification (Desboefus et al., 2005; Oakes et al., 2012a). Therefore, it seems that coal-derived $SO_2$ or $H_2SO_4$ emitted with Pb by coal combustion solubilizes Fe, resulting in the correlation between $Fe_{sol}$% and EF of Pb that may have occurred with nss-$SO_4^{2-}$ as a mediator variance.

Although $PM_{1.3}$ in the CPO sample did not pass over the highly polluted region, S6-CPO had a positive $[H^+]_{mineral}$ and high $Fe_{sol}$%. The $[H^+]_{mineral}$ was derived from $SO_2$ and $H_2SO_4$ because the nss-$SO_4^{2-}$ concentration in S6-CPO was approximately an order of magnitude higher than that of $NH_4^+$ (Figs. S3b and S3c); similar results have been reported in previous studies (Paulot et al., 2015; Nault et al., 2021). The possible sources of $SO_2$ and $H_2SO_4$ in the Pacific Ocean in the Southern Hemisphere are mainly biogenic S (e.g., dimethyl sulfide), which is indicated by the S isotope ratio (Calhoun et al., 1991; Li et al., 2018). Moreover, it is known that mineral dust is internally mixed with sulfate through cloud processes, even if it does not pass over the polluted region (Fitzgerald et al., 2015). Therefore, the mineral dust in the CPO samples was likely acidified by biogenic S during transport.

### 3.3. Size dependence of Fe species in marine aerosols

Iron species in $PM_{>1.3}$ were composed of two or three Fe species: hematite, ferrihydrite, biotite, and illite (Figs. 7a, 7b, and S4). More than half of the Fe in $PM_{1.3}$ was composed of biotite (Fig. 7a). The EXAFS spectra of $PM_{>1.3}$ accurately reflected the spectrum feature of biotite in 7–9 Å of k-space (Figs. S5a–S5c), indicating that biotite was the dominant Fe species at $PM_{>1.3}$. The relative abundance of ferrihydrite increased with decreasing aerosol diameter and increasing transportation distance (Fig. 7a, transport distance: WPO1 < WPO2 < WPO3 < SPO $\cong$ CPO). The hydration reaction of phyllosilicates in $PM_{>1.3}$ forms secondary ferrihydrite during transportation (Takahashi et al., 2011). Therefore, Fe in biotite at $PM_{>1.3}$ was partially altered to ferrihydrite. The Fe species in $PM_{1.3}$ with negative $[H^+]_{mineral}$ (S5-WPO1 and S6-SPO), were composed of the same species in $PM_{>1.3}$ (Figs. 6a and 7a). The negative $[H^+]_{mineral}$ value indicates that the mineral dust was not acidified beyond the buffering capacity of $CaCO_3$. Therefore, the Fe species in $PM_{>1.3}$ and $PM_{1.3}$ were not drastically modified by aerosol acidification.

Iron(III)-HULIS and Fe(III)-sulfate were found as characteristic Fe species in $PM_{1.3}$ with $[H^+]_{mineral}$ and high $Fe_{sol}$% (>10 %, Figs. 7a, 7b, and S4). Iron(III)-HULIS was present in all $PM_{1.3}$ with positive $[H^+]_{mineral}$, whereas only S6-WPO3 contained Fe(III)-sulfate and Fe(III)-HULIS (Fig. 7a). Since the source of Fe in $PM_{1.3}$ was mineral dust, the Fe species at the time of emission was thought to be similar to that of $PM_{>1.3}$. However, the EXAFS spectra of $PM_{1.3}$ reflected spectrum features of Fe(III)-HULIS and Fe(III)-sulfate rather than biotite (Fig. S5). It appears that Fe(III)-sulfate and Fe(III)-HULIS were formed by secondary processes of biotite during transport. Oxalate is an important ligand for enhancing $Fe_{sol}$% in aerosol particles (Chen and Grassian, 2013; Ito and Shi, 2016; Hamilton et al., 2019), and the presence or absence of Fe(III)–

oxalate in these samples was examined. As a result, the abundance of Fe(III)–oxalate in these samples was not the dominant Fe species in our samples obtained by LCF. This result is consistent with the fact that there was no correlation between the $Fe_{sol}$% and oxalate concentrations (Spearman's $\rho$ =0.20).

For comparison, the Fe species in East Asian aerosols (Beijing and NOTOGRO) were determined by XAFS spectroscopy. The EF of Fe and $Fe_{sol}$% in Beijing dust were 0.85 and 0.53 % (MQ extraction at 100 g/L of the dust/liquid ratio), respectively. Although the sampling year of the sample collected in NOTOGRO was different from that of the marine aerosol particles, the backward trajectory, EF of Fe, and $Fe_{sol}$% in the sample collected in the sample were similar to those of the marine aerosol particles (Fig. S6a–S6d). Therefore, this sample was used as a proxy for the chemical alteration of Fe in size-fractionated aerosol particles during transport from East Asia to Japan. Beijing dust also contained Fe(II)-sulfate and Fe(III)-sulfate with ferrihydrite and biotite. Relative abundances of these species to total Fe were 9 %, 11 %, 44 %, and 36 %, respectively (Fig. S4d). The iron species in $PM_{>1.3}$ collected in NOTGORO, were composed of illite, smectite, biotite, and ferrihydrite, the species of which were similar to those in $PM_{>1.3}$ in marine aerosol particles (Figs. 7a and 7c), whereas Fe(II)-sulfate and Fe(III)-oxalate were found in $PM_{1.3}$ collected in NOTOGRO (Figs. 7a and 7c). The EXAFS spectrum of S6-NT has a single peak in 7–9 Å of k-space, whereas those of $PM_{>1.3}$ has two peaks of biotite in the same regions (Fig. S5d). Therefore, Fe(II)-sulfate and Fe(III)-oxalate were formed by the chemical alteration of biotite, which is consistent with the Fe speciation results of WPO. Iron(III)-HULIS was not identified as the dominant Fe species in NOTOGRO and Beijing dust (Figs. 5a and S5d). These results indicate that Fe(III)-HULIS in the WPO samples was possibly formed by the chemical alteration of Fe(II, III)-sulfates and Fe(III)-oxalate after aerosol passes over Japan.

To identify the L-Fe species in $PM_{1.3}$, Fe K-edge XANES spectra of insoluble Fe in S6-WPO2 were recorded after the water extraction of labile Fe species. The XANES spectra of total Fe (labile + insoluble Fe) were well fitted by insoluble Fe and Fe(III)-HULIS (Fig. S4e). Furthermore, the $Fe_{sol}$% in $PM_{1.3}$ was correlated with the abundance of Fe(III)-HULIS (Fig. 7d). Therefore, Fe(III)-HULIS is an important L-Fe species in $PM_{1.3}$ in the marine atmosphere.

### 3.4. Size dependence of Al species

Given that Fe(III)-HULIS was formed by chemical alterations of Fe in biotite, the Al species in $PM_{1.3}$ may be different from those in $PM_{>1.3}$. Therefore, we determined the Al species in WPO2 and WPO3 by XANES spectroscopy. The Al species in $PM_{>1.3}$ were composed of octahedral Al and tetrahedral Al, of which the Al K-edge XANES spectra were similar to those of biotite (Fig. 8). Moreover, the Al K-edge XANES spectra of $PM_{>1.3}$ were well fitted by each other. This result implies that (i) the dominant Al species in $PM_{>1.3}$ were 2:1 phyllosilicate, including biotite, and (ii) Al species in these samples did not change significantly during transport. In contrast, secondary Al species were identified in the $PM_{1.3}$. Gibbsite was found in S5-WPO2 and S5-WPO3, with which abundances of 20 % and 30 % in total Al, respectively (Figs. 8a and 8b). The S6-WPO3 contained Al-sulfate and organic complexes of Al (organo-Al), gibbsite, and phyllosilicates, and their abundances were 8 %, 8 %, 18 %, and 66 %, respectively. The presence of organo-Al and Al-sulfate in S6-WPO3 is consistent with that of Fe(III)-sulfate in this sample (Figs. 7a and 8b). In the case of S6-WPO2, the Al K-edge XANES

spectrum was completely different from that of phyllosilicates at $PM_{>1.3}$ (Fig. 8a), although the XANES spectrum of S6-WPO2 could not be fitted by the reference materials examined in this study. Given that the initial Al species in $PM_{1.3}$ were phyllosilicates, as was the case for $PM_{>1.3}$, it is possible that phyllosilicate particles were altered in the atmosphere. This result is consistent with the absence of the spectral feature of biotite in the Fe K-edge EXAFS spectra of $PM_{1.3}$.

## 3.5. Single-particle analysis

Single-particle analysis of S6-WPO2 was conducted using STXM for evaluating the alteration processes of Fe-bearing phyllosilicate particles. Iron-bearing particles had irregular shapes (Figs. S7 and S8). This morphological feature is similar to that of naturally occurring phyllosilicate particles (Matsuki et al., 2005; Jeong and Nousiainen, 2014). In contrast, anthropogenic Fe (e.g., fly ash and pyrogenic hematite) has spherical shapes that are not dominant in S6-WPO2 (Li and Shao, 2009; Adachi et al., 2021). Therefore, Fe-bearing particles with irregular shapes were phyllosilicate particles. These Fe-bearing phyllosilicates are covered with Na and OCs. The carbon K-edge NEXAFS spectra on the surface of mineral dust were similar to those of OCs in submicron aerosol particles reported in previous studies (Prather et al., 2013; Wilson et al., 2015). Furthermore, the Na species on the particle surface were similar to the Na salt with organic acids rather than inorganic Na (Figs. S8 and S9d), for which the Na K-edge XANES spectra were similar to the average Na species in $PM_{1.3}$ collected in WPO2 and WPO3 (Figs. S9a and S9b). Submicron SSA and marine cloud water contain both Na and OCs (Mochida et al., 2002; Straub et al., 2007; Cochran et al., 2016; Bikkina et al., 2019). Therefore, it is considered that the mineral dust gained Na and OCs on the particle surface through cloud processes.

Similar internal mixing particles between mineral dust and SSA have been found not only in the Pacific Ocean, but also in other regions (Okada et al., 1990; Niimura et al., 1998; Wagner et al., 2008; Kandler et al., 2017; Adachi et al., 2020; Kwak et al., 2022; Knopf et al., 2022). It is considered that the internal mixing of mineral dust and sea salt is formed by cloud processes (Niimura et al., 1998; Formenti et al., 2011). A recent study found internal mixing particles between aged sea salt, mineral dust, S, and OCs in submicron aerosols collected from the North Atlantic Ocean, of which OCs species were similar to those in SSA (Knopf et al., 2022). This result is similar to the microscopic observation results (Figs. S7 and S8). Since (i) SSA is ubiquitously present in the marine atmosphere and (ii) the chemical composition of marine cloud water is influenced by SSA, the internal mixing of mineral dust with SSA in cloud water may play a significant role in the organic complexation of L-Fe in aerosol particles in the marine atmosphere.

## 4. Discussion

### 4.1. Reconstruction alteration processes of Fe based on $pH_{PPD}$ and $pH_{L\text{-}Fe}$

Our results showed that L-Fe in aerosol particles was mainly controlled by Fe in $PM_{1.3}$ (Fig. 4c). Aerosol acidification was one of the factors of enhancement of L-Fe concentrations because (i) $PM_{1.3}$ with high $Fe_{sol}\%$ (>10 %), has positive $[H^+]_{mineral}$ (Fig. 6a), and (ii) L-Fe concentration correlated with $[nss\text{-}SO_4^{2-}]$ (Fig. 5d). Furthermore, Fe(III)-HULIS was found in $PM_{1.3}$ with a positive $[H^+]_{mineral}$, of which the fraction of Fe(III)-HULIS correlated with $Fe_{sol}\%$ in aerosol particles (Fig.

7d). Therefore, it is likely that both aerosol acidification and organic complexation of Fe contributed to the enhancement of Fe$_{sol}$%. The reaction pH for proton-promoted dissolution (pH$_{PPD}$) and formation of L-Fe species (pH$_{L-Fe}$) were evaluated using conceptual and geochemical models, respectively. First, the modeled L-Fe concentration in PM$_{1.3}$ with a negative [H$^+$]$_{mineral}$ was much higher than the observed L-Fe concentration, even though pH$_{PPD}$ was set as 3.0 (Fig. S10). This result indicates that the Fe-bearing particles in these samples were not acidified to a pH of 3.0. Consequently, Fe in these samples was not sufficiently solubilized by atmospheric processes. In fact, the Fe species in these samples were similar to those in PM$_{>1.3}$ with low Fe$_{sol}$%.

The observed L-Fe concentrations in PM$_{1.3}$ with positive [H$^+$]$_{mineral}$ were reproduced when the pH was < 3.0 (Figs. 9a, 9c, and S11). This result is consistent with those of previous studies because a high Fe$_{sol}$% was observed when the aerosol pH was lower than 3.0 (Fang et al., 2017; Tao and Murphy, 2019). As previously mentioned, the Fe in the biotite was altered to Fe(III)-HULIS and/or Fe(III)-sulfate. Previous studies have shown that the octahedral layer of phyllosilicates, including biotite, is preferentially decomposed under highly acidic conditions (pH < 3.0), and Fe in biotite is mainly present in the octahedral layer (Shaw et al., 2009; Bray et al., 2015). Therefore, it is reasonable that Fe$_{sol}$% increased rapidly when the pH dropped below 3.0. The modeled L-Fe species in PM$_{1.3}$ with positive [H$^+$]$_{mineral}$ were present as Fe(III)-sulfate or Fe(III)-oxalate under acidic conditions (pH$_{L-Fe}$ < 3.0) with any ratio of [citrate]/[oxalate] and [citrate]/[L-Fe], although the stability constants of citrate are much higher than those of oxalate and sulfate (Figs. 9b, 9d, and S12). This phenomenon can be ascribed to the fact that citric acid forms fully protonated species below its pK$_{a1}$ (= 3.13), whereas oxalate and sulfate can form ferric complexes, even at pH < 3.0 (Figs. 9b, 9d, and S12). As previously mentioned, the East Asian aerosol particles contained Fe(II, III)-sulfate and Fe(III)-oxalate, but Fe(III)-HULIS was not the dominant Fe species (Figs. 6a and S4d). Therefore, it is considered that the mineral dust in the WPO samples encountered highly acidic conditions during transportation in East Asia. By contrast, the CPO sample did not pass over the polluted region (Fig. S1b), and positive [H$^+$]$_{mineral}$ and low pH$_{PPD}$ were observed in S6-CPO (Figs. 6a and S11a). Aluminosilicate particles react with sulfate through cloud processes, even if the particles do not pass over the polluted region (Fitzgerald et al., 2015). Moreover, a previous study reported that the Fe$_{sol}$% in Saharan dust was increased by aerosol acidification by nss-SO$_4^{2-}$ during long-range transport in the Atlantic Ocean (Longo et al., 2016). Therefore, similar reaction processes can promote the acidification of the CPO sample. Although nss-SO$_4^{2-}$ in the pelagic region is thought to be derived from biogenic origins (Calhoun et al., 1991; Li et al., 2018), further studies are required for determining the effect of biogenic S on the increase in Fe$_{sol}$%.

In contrast, the geochemical model showed that Fe(III)-HULIS was dominant under moderately acidic conditions (pH$_{L-Fe}$ 3.0–6.0), where the [citrate]/[L-Fe] ratio was higher than 1.0 (Figs. 9b, 9d, and S12). In S6-WPO3, the coexistence of Fe(III)-HULIS and Fe(III)-sulfate was found only under moderately acidic conditions, when [citrate]/[L-Fe] was between 0.30 and 0.45 (Figs. 9d). Therefore, the pH of phyllosilicates should be increased after proton-promoted processes to form Fe(III)-HULIS. Single-particle analyses identified the presence of a surface coating of Na and OCs on phyllosilicate particles, which was caused by internal mixing with submicron SSA or marine cloud water (Fig. S7 and S8). A recent mesocosm experiment showed that submicron SSA is rapidly acidified to pH 2.0, because of water evaporation, uptake of acidic gases,

and/or displacement of protons in organic acids by $Na^+$ (Angle et al., 2021). Our Na speciation results showed that the organic salt of Na was present in the submicron SSA (Fig. S9). If submicron SSA in the ambient atmosphere has high aerosol acidity, the internal mixing of phyllosilicates and submicron SSA may not sufficiently increase the pH of phyllosilicates.

Another potential process for increasing aerosol pH is the evaporation–condensation cycle (cloud process) during transportation. Marine cloud water can also form Na and OCs coatings on phyllosilicate particles because Na and OCs are dominant components in marine cloud water (Straub et al., 2007). Given that 500 nm of phyllosilicate particles with a 100 nm-thick water layer at pH 1.0 was incorporated into typical marine cloud water (diameter 10 μm, pH 4.0, Boris et al., 2016; Kim et al., 2019; Shah et al., 2020), the pH of aerosol particles was 3.97. The increase in aerosol pH by cloud processes decreases $Fe_{sol}$% because of the precipitation of nano-ferrihydrite, with the sole consideration of inorganic Fe chemistry (Spokes et al., 1994; Shi et al., 2015; Maters et al., 2016). However, nano-ferrihydrite precipitation was suppressed by the formation of Fe(III)-HULIS owing to its high solubility. As a result, L-Fe obtained by proton-promoted dissolution below pH 2.0 was retained under moderately acidic conditions. Therefore, the role of HULIS is not to encourage further Fe dissolution from aerosol particles, but to stabilize L-Fe under moderately acidic conditions. This result was consistent with that of a previous laboratory experiment (Paris and Desboeufs, 2013).

In summary, Fe in $PM_{1.3}$ was solubilized by proton-promoted dissolution, and subsequently, solubilized Fe was stabilized as L-Fe by organic complexation with HULIS in the cloud processes (Fig. 10). In the case of the WPO samples, aerosol acidification and stabilization of L-Fe occurred in East Asia and the Pacific Ocean, respectively. This result is consistent with the hypothesis proposed by Buck et al. (2013). These studies imply that atmospheric processing after passing over Japan is not important for solubilizing Fe because significant differences in $Fe_{sol}$% in the North Pacific Ocean have not been observed (Buck et al., 2013). The stabilization of L-Fe species may play a critical role in the supply of dissolved Fe from aerosol particles to the ocean surface. Given that log $K_{HULIS}$ in aerosol particles is a strong ligand on the ocean surface ($L_1$, log K >12), Fe(III)-HULIS dissolves without the consumption of $L_1$ ligands on the ocean surface. This phenomenon possibly promoted further Fe dissolution with moderately water-soluble species (e.g., nano-ferrihydrite) by complexation with $L_1$ or weaker ligands ($L_2$, log K:11–12) on the ocean surface (Gledhill and Buck, 2012). When log $K_{HULIS}$ was similar to weak or super-weak ligands (log K < 11), the probability of encountering $L_1$ and $L_2$ ligands with Fe(III)-HULIS increased with the expanding lifetime of dissolved Fe (hours to days, Meskhidze et al., 2017). Thus, Fe(III)-HULIS strongly influences the fate of dissolved Fe in the ocean from the aerosol particles. Further investigation of atmospheric organic ligands combined with Fe in aerosol particles is necessary for gaining further knowledge of the biogeochemical cycle of Fe.

**4.2. Importance of size-fractionated aerosol particles**

Thus, the Fe in $PM_{1.3}$ was solubilized by atmospheric processes during transportation. These results could not be obtained if we collected, rather than size-fractionated aerosol particles. This is because the abundance of Fe(III)-HULIS is

approximately 5 % of total Fe in TSP, which is below the detection limit of XAFS spectroscopy. Previous studies have also suggested the presence of Fe(III)-sulfate as an L-Fe species by spot analysis using microscopic XAFS, but Fe(III)-sulfate was not detected by macroscopic XAFS because of the lower abundance of the species in TSP (Oakes et al., 2012b; Kurisu et al., 2021). Therefore, size-fractionated aerosol sampling is required to identify the L-Fe species in marine aerosol particles.

In general, the cut-off diameter for size-fractionated aerosol sampling is 2.5 μm, but this may not be sufficient to separate the L-Fe species with high $Fe_{sol}$% from the less aged mineral dust. Our results showed that a high $Fe_{sol}$% associated with Fe(III)-HULIS was found in $PM_{1.3}$. In contrast, aerosol particles in stage-4 ($PM_{1.0-2.5}$) did not have a high $Fe_{sol}$% because mineral dust in the fraction was not acidified because of the negative $[H^+]_{mineral}$. This result is consistent with previous studies because the aerosol pH in $PM_{1.0-2.5}$ was higher than that in $PM_{1.0}$, owing to the larger and smaller

contributions of non-volatile cations (e.g., Ca and Na) and sulfates in $PM_{1.0-2.5}$ compared to $PM_{1.0}$, respectively (Fang et al., 2017; Guo et al., 2018). Furthermore, in our sample, approximately 40 % (11.9–58.9 %) of the total Fe in $PM_{2.5}$ was contained in $PM_{1.0-2.5}$. In the analysis of $PM_{2.5}$, the relative abundances of L-Fe concentrations in $PM_{1.0}$ were diluted by insoluble Fe in $PM_{1.0-2.5}$. This is also relevant to the investigation of pyrogenic Fe with high $Fe_{sol}$%. Previous studies have shown that a low Fe isotope ratio associated with pyrogenic Fe is found in $PM_{1.3}$, whereas the isotope ratio in $PM_{1.0-2.5}$ is

similar to that of Fe in crustal materials (Kurisu et al., 2016, 2019). For these reasons, two-stage aerosol sampling with a cut-off diameter of 1.0 μm or multi-stage aerosol sampling is desirable for investigating the factors controlling $Fe_{sol}$% in marine aerosol particles. Because size-fractionated aerosol sampling recovers a small amount of sample per stage compared to TSP sampling (Sakata et al., 2018; Baker et al., 2020), the development of analytical techniques for low concentrations of trace metals is essential.

## 5.    Conclusions

    In this study, size-fractionated aerosol particles were collected in the Pacific Ocean. About 80 % of total Fe were present in $PM_{>1.3}$, whereas $PM_{1.3}$ accounted for about 60 % of L-Fe in TSP. The average $Fe_{sol}$% in $PM_{1.3}$ (22.3±21.7 %) was about an order magnitude of higher than that in $PM_{>1.3}$ (2.56±2.53 %). The Fe species in $PM_{>1.3}$ were ferrihydrite, hematite,

biotite, and illite. These Fe were similar to those in mineral dust. The Fe in $PM_{>1.3}$ was not well solubilized during transportation because mineral dust in the fraction was not acidified beyond the buffering capacity of $CaCO_3$. In the case of $PM_{1.3}$ with positive $[H^+]_{minerals}$, Fe(III)-HULIS was present as specific L-Fe species in $PM_{1.3}$. The species were formed by the chemical alteration of biotite. The chemical alteration of biotite in $PM_{1.3}$ was confirmed by bulk Al speciation and single-particle analysis of mineral dust because secondary Al species (e.g., gibbsite, Al-sulfate, and organo-Al) were present in the

$PM_{1.3}$. Thus, the Al species can be used as a good indicator of the degree of phyllosilicate alterations. As a result of pH estimation using a conceptual model, Fe in mineral dust was solubilized under highly acidic conditions ($pH_{PPD} < 3.0$). Subsequently, Fe(III)-HULIS was formed in marine cloud water under moderately acidic conditions ($3.0 < pH_{L-Fe} < 6.0$). The role of the complexation reaction of Fe with HULIS is the stabilization of L-Fe rather than the further promotion of Fe

dissolution from aerosol particles. At present, thermodynamic data of HULIS with Fe in $PM_{1.3}$ are not enough to evaluate the effects of HULIS on Fe dissolution. Therefore, further observations and laboratory experiments on the complex formation between HULIS and Fe are expected to improve our understanding of the effect of HULIS on Fe dissolution.

*Data availability*. The data are available upon request (Kohei Sakata, sakata.kohei@nies.go.jp).

*Supplement.* The supplement related to this article is available online at XXXX.

*Author contributions.* The study was designed by Kohei Sakata (K.S.), Hiroshi Tanimoto (H.T.), and Yoshio Takahashi (Yo.T.). Aerosol sampling was conducted by K.S., Aya Sakaguchi (A.S.), and Atsushi Matsuki (A.M.). Quantitative analyses were conducted by K.S., Minako Kurisu (M.K.), and Yo.T. XAFS experiments were conducted by K.S., M.K., Yo.T., Yusuke Tamenori (Yu. T.), and Yasuo Takeichi (Ya.T.). Single particle analyses were performed by K.S., Ya.T., and Yo. T. The model calculations were performed by K.S. The paper was written by K.S., H.T., and Yo.T. All authors were reviewed the manuscript.

*Competing interests*. The authors declare no competing interests.

*Acknowledgements.*

We thank all researchers and clues of KH-14-6 cruise to support our aerosol sampling. This study was supported by a Grant-in-Aid for the Japan Society for the Promotion of Science (JSPS) Fellows (Proposal No. 201801782), Research Institute for Oceanochemistry Foundation (Kyoto, Japan, Proposal No. H30-R4), and the Cooperative Research Program of the Institute of Nature and Environmental Technology, Kanazawa University (Proposal No. 19002). XAFS experiments were approved by KEK-PF (2013S2-003, 2015S2-002, 2016G632, 2018S1-001, 2019G093) and SPring-8 (2015A1809 and 2016A1642).

**Figure captions**

Figure 1: Track chart of the research cruise of KH-14-6 (R/V Hakuho-Maru) and sampling locations of WPO, CPO and SPO samples. Red circle showed the locations of Beijing and Noto Ground-Based Research Observatory (NOTOGRO). The figure was described using Ocean Data View (Schlitzer, 2021).

Figure 2: Iron K-edge (a) XANES and (b) EXAFS spectra of reference materials.

Figure 3: (a) total Fe ($ng/m^3$), (b) labile Fe ($ng/m^3$), (c) $Fe_{sol}$%, (d) EF of Fe, (e) total Al ($ng/m^3$), (f) labile Al ($ng/m^3$), and (g) $Al_{sol}$% in TSP.

Figure 4: Size distributions of (a) total Fe ($ng/m^3$), (b) labile Fe ($ng/m^3$), (c) $Fe_{sol}$%, (d) total Al ($ng/m^3$), (e) labile Al ($ng/m^3$), and (f) $Al_{sol}$%. The $PM_{1.3}$ is shown in yellow regions.

Figure 5: Scatter plots of $Fe_{sol}$% with (a) EF of V, (b) EF of Ni, (c) EF of Pb, and (d) nss-$SO_4^{2-}$. (e) the scatter plot between nss-$SO_4^{2-}$ and EF of Pb.

Figure 6: (a) A scatter plot between $Fe_{sol}$% and $[H^+]_{mineral}$. The blue region shows positive $[H^+]_{mineral}$. Size distributions of (b) $[H^+]_{mineral}$ and (c) nss-$Ca^{2+}$. The $PM_{1.3}$ is shown in yellow regions.

Figure 7: (a) Fraction of Fe species in each sample determined by Fe K-edge XANES spectroscopy. Iron K-edge XANES spectra of size-fractionated aerosol particles collected in (b) WPO2 and (c) NOTOGRO. (d) a scatter plot between fraction of Fe(III)-HULIS and $Fe_{sol}$% in $PM_{1.3}$.

Figure 8: Al K-edge XANES spectra of (a) WPO2 and (b) WPO3. Black and red solid line showed XANES spectra for aerosol particles and fitting spectra, respectively. Colored spectra with dashed line show fitting components. The relative abundance of species identified by LCF are shown in the parentheses beside the sample name (i.e., Gibbsite (20%) for S5-WPO2).

Figure 9: (a and c) dissolution curves for each Fe pool (colored dashed lines) and summation of all Fe pools (solid black line) in S6-WPO2 and S6-WPO3 as a function of dissolution time. Solid red line in these figures shows the observed L-Fe concentrations. The pH was set so that the total value reached the observed L-Fe in approximately 90 h (expected time for wet aerosol phase). (b and d) pH dependences of L-Fe species in ALW for S6-WPO2 and S6-WPO3. Pink and yellow regions show the aerosol pH for the proton-promoted dissolution (same pH as in panels a and c) and stable pH regions of Fe(III)-HULIS, respectively.

Figure 10: The schematic of alteration processes of Fe in phyllosilicate particles in $PM_{1.3}$ during transport.

**Figures**

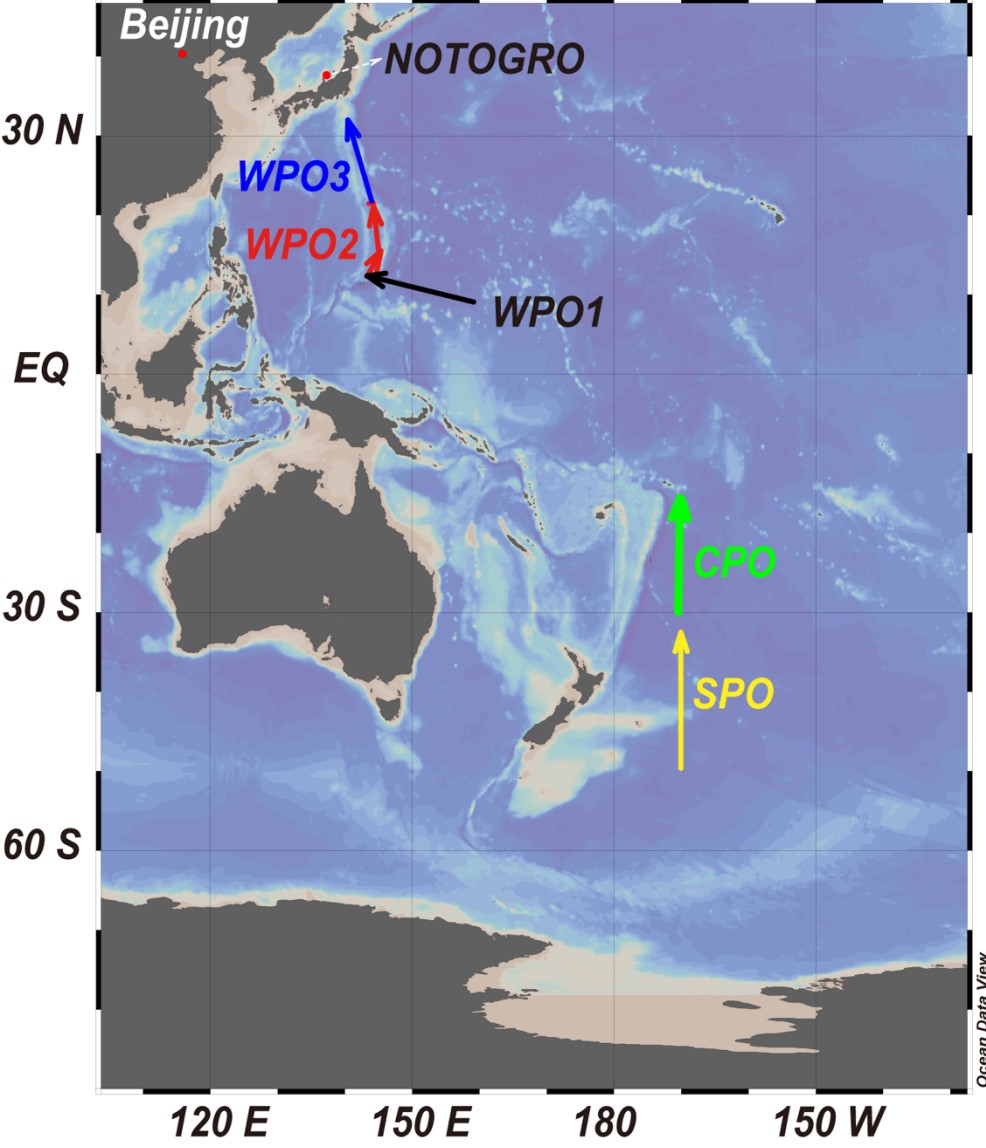

Figure 1: Track chart of the research cruise of KH-14-6 (R/V Hakuho-Maru) and sampling locations of WPO, CPO and SPO samples. Red circle showed the locations of Beijing and Noto Ground-Based Research Observatory (NOTOGRO). The figure was described using Ocean Data View (Schlitzer, 2021).

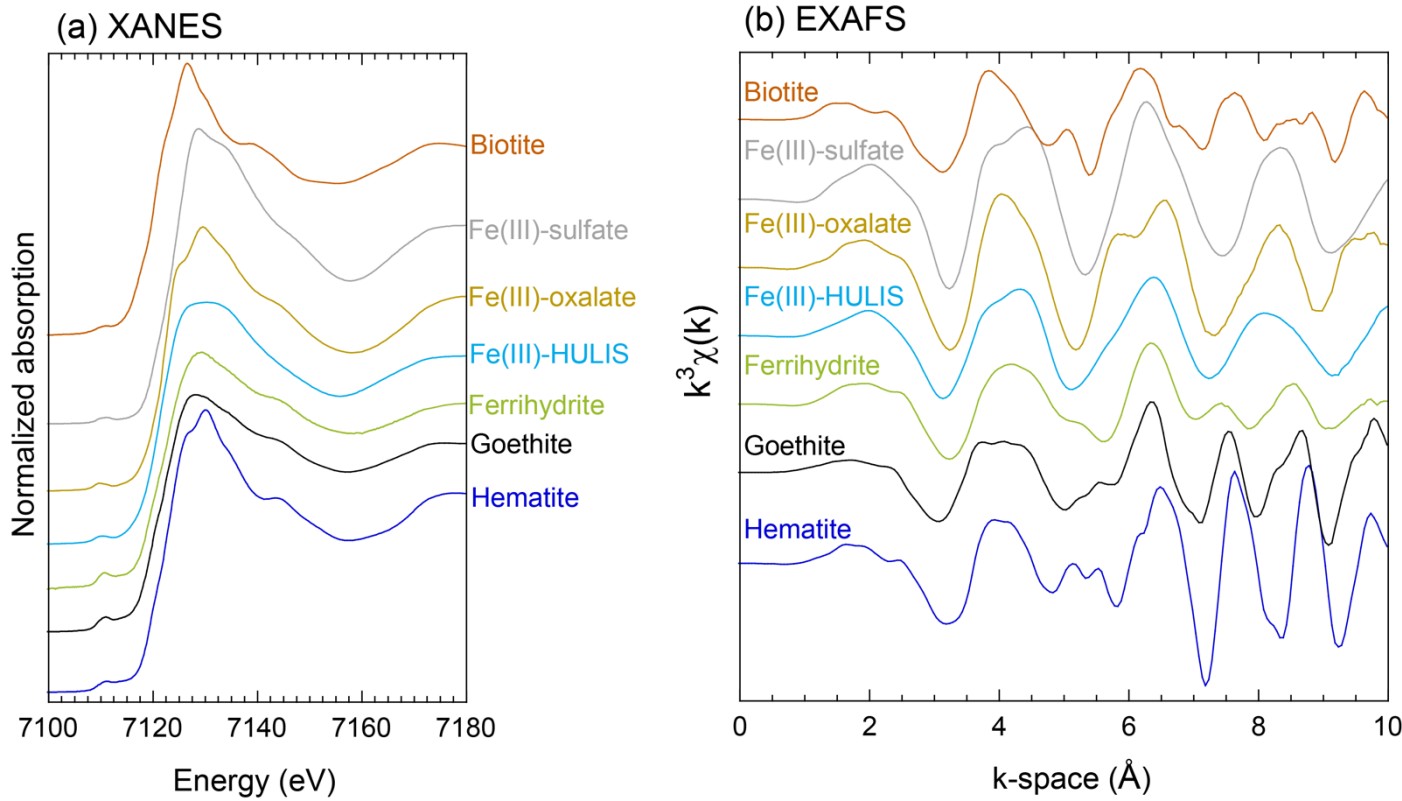

Figure 2: Iron K-edge (a) XANES and (b) EXAFS spectra of reference materials.

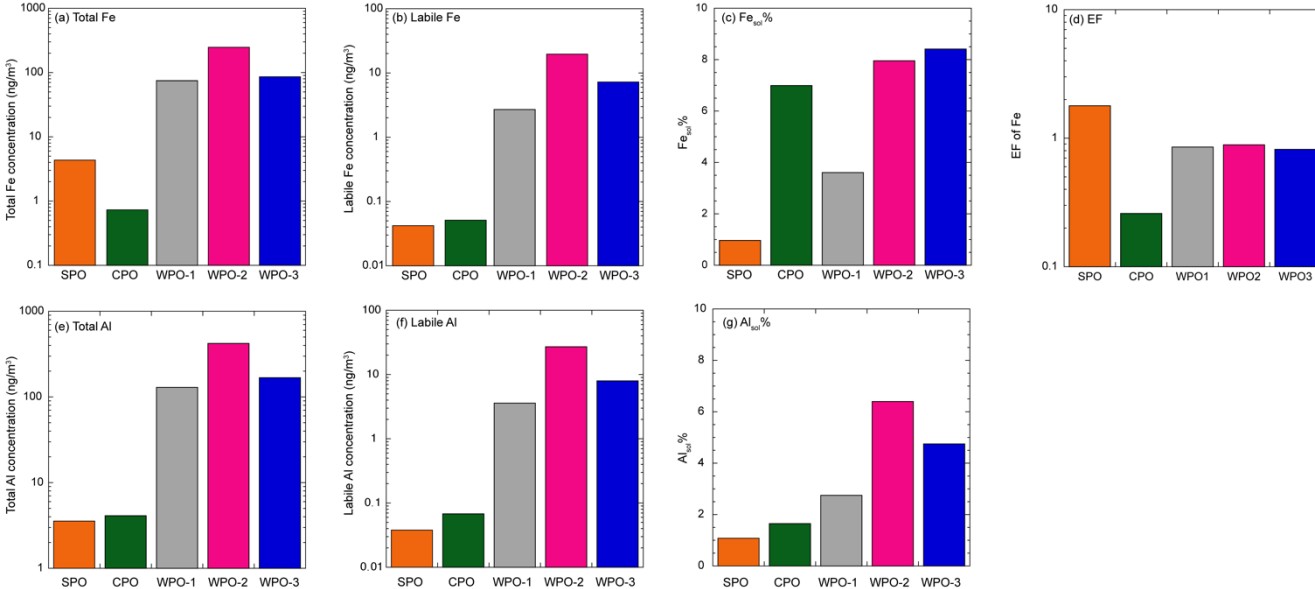

Figure 3: (a) total Fe (ng/m$^3$), (b) labile Fe (ng/m$^3$), (c) Fe$_{sol}$%, (d) EF of Fe, (e) total Al (ng/m$^3$), (f) labile Al (ng/m$^3$), and (g) Al$_{sol}$% in TSP.

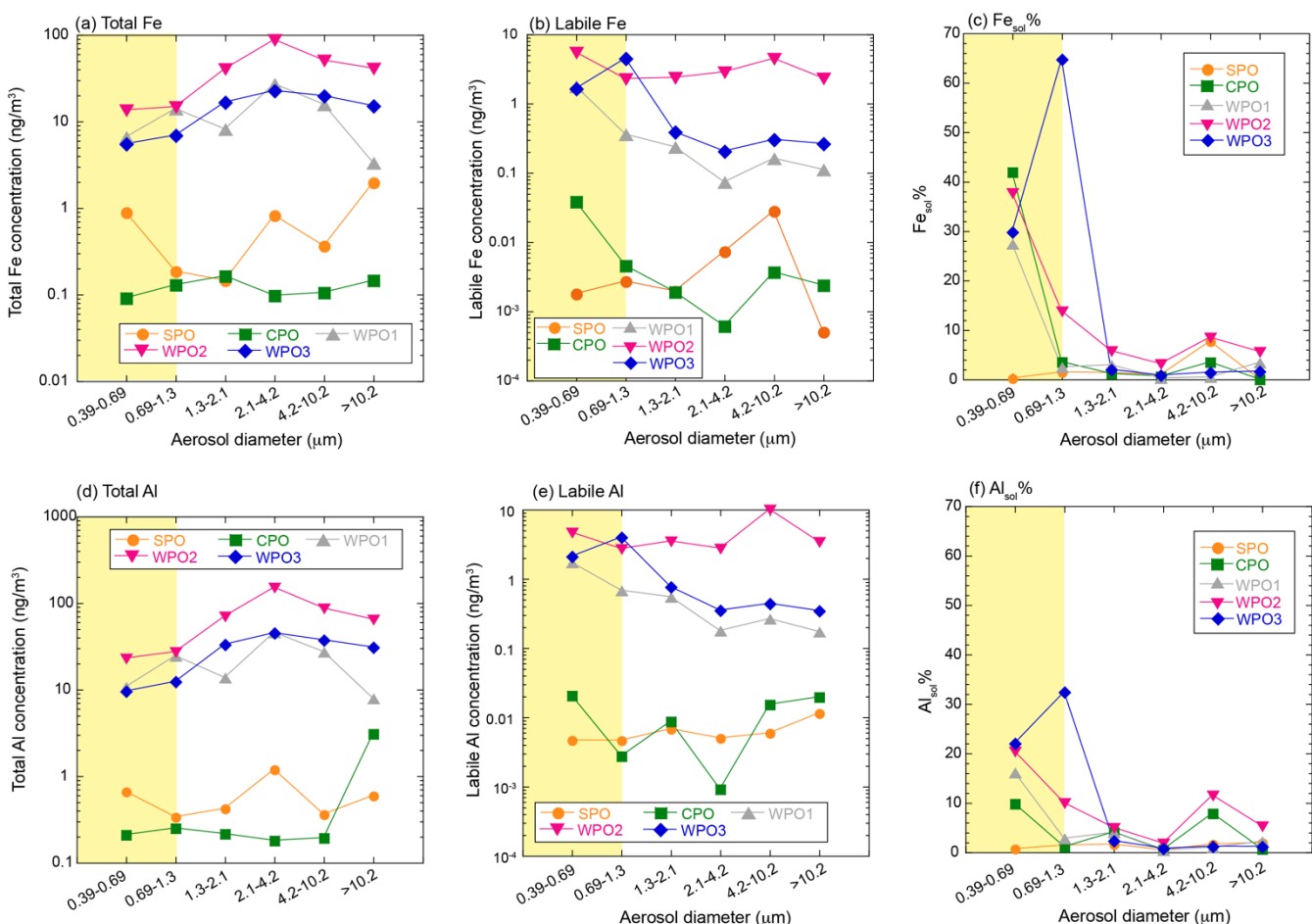

Figure 4: Size distributions of (a) total Fe (ng/m$^3$), (b) labile Fe (ng/m$^3$), (c) Fe$_{sol}$%, (d) total Al (ng/m$^3$), (e) labile Al (ng/m$^3$), and (f) Al$_{sol}$%. The PM$_{1.3}$ is shown in yellow regions.

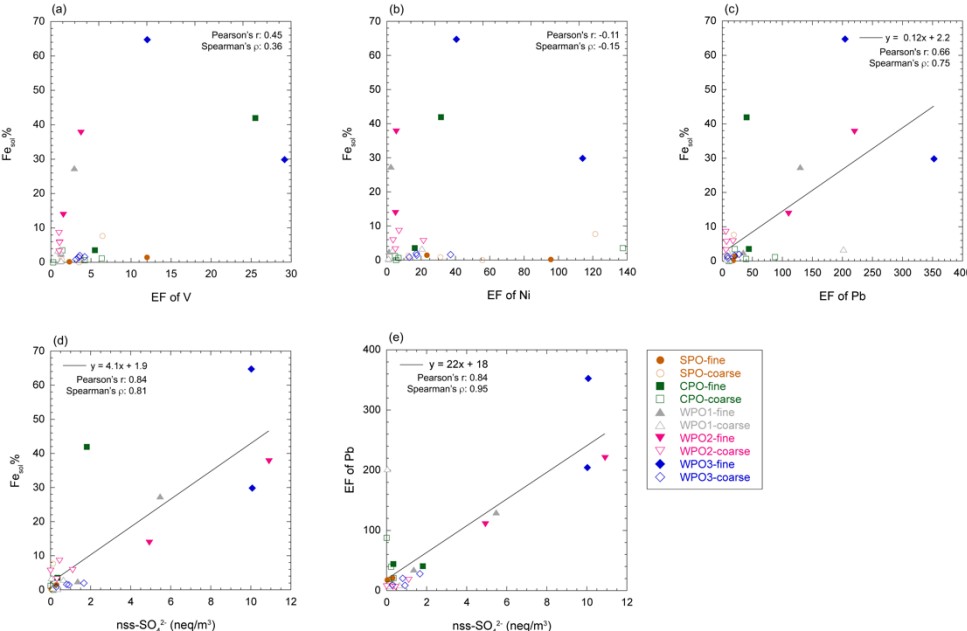

Figure 5: Scatter plots of $Fe_{sol}\%$ with (a) EF of V, (b) EF of Ni, (c) EF of Pb, and (d) nss-$SO_4^{2-}$. (e) the scatter plot between nss-$SO_4^{2-}$ and EF of Pb.

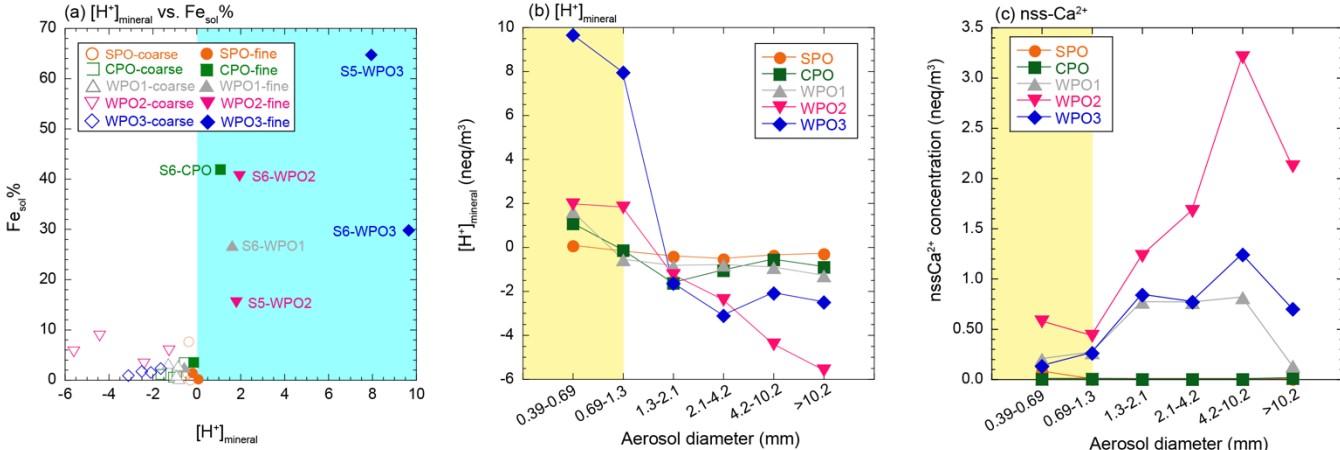

Figure 6: (a) A scatter plot between $Fe_{sol}\%$ and $[H^+]_{mineral}$. The blue region shows positive $[H^+]_{mineral}$. Size distributions of (b) $[H^+]_{mineral}$ and (c) nss-$Ca^{2+}$. The $PM_{1.3}$ is shown in yellow regions.

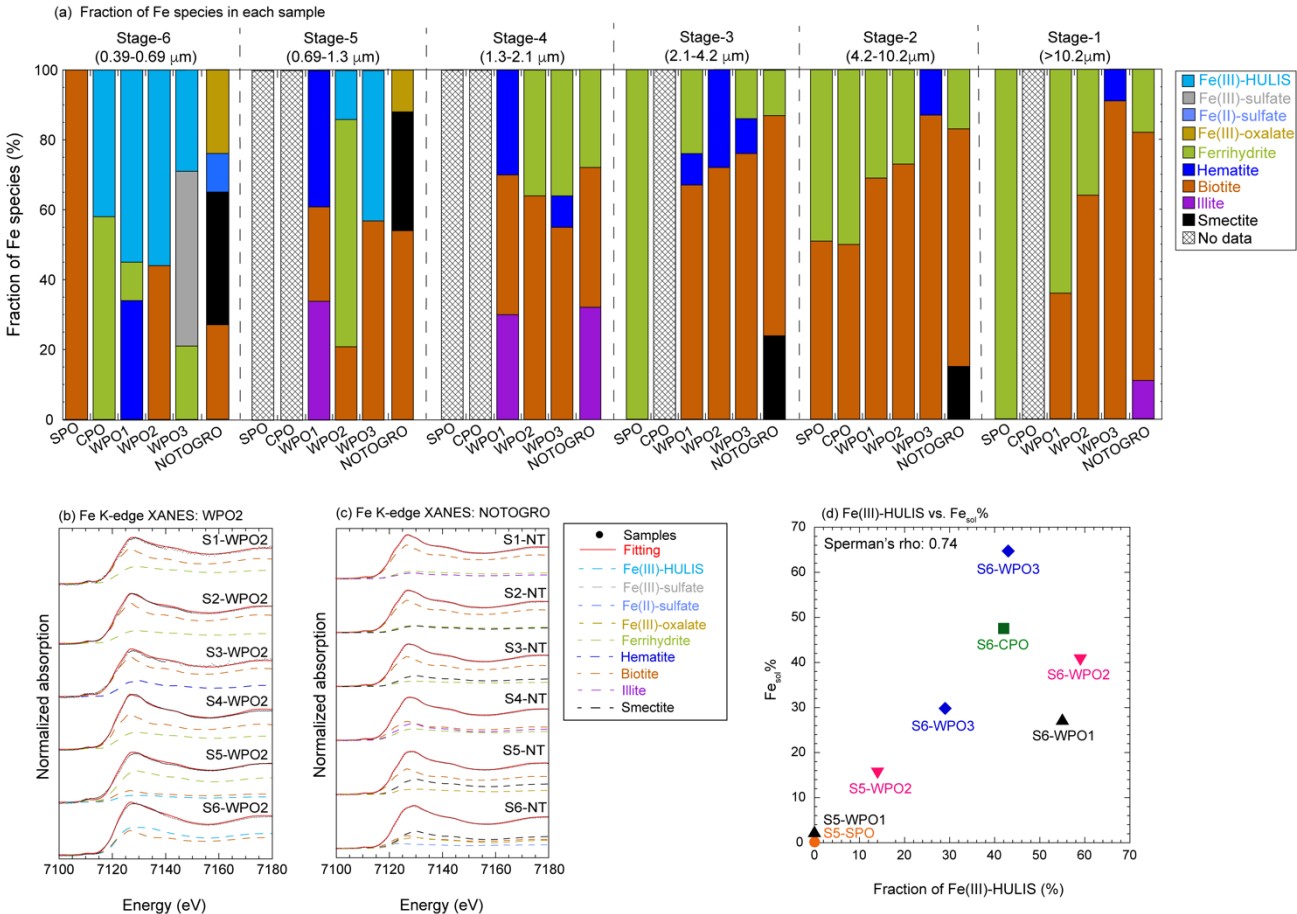

Figure 7: (a) Fraction of Fe species in each sample determined by Fe K-edge XANES spectroscopy. Iron K-edge XANES spectra of size-fractionated aerosol particles collected in (b) WPO2 and (c) NOTOGRO. (d) a scatter plot between fraction of Fe(III)-HULIS and $Fe_{sol}$% in $PM_{1.3}$.

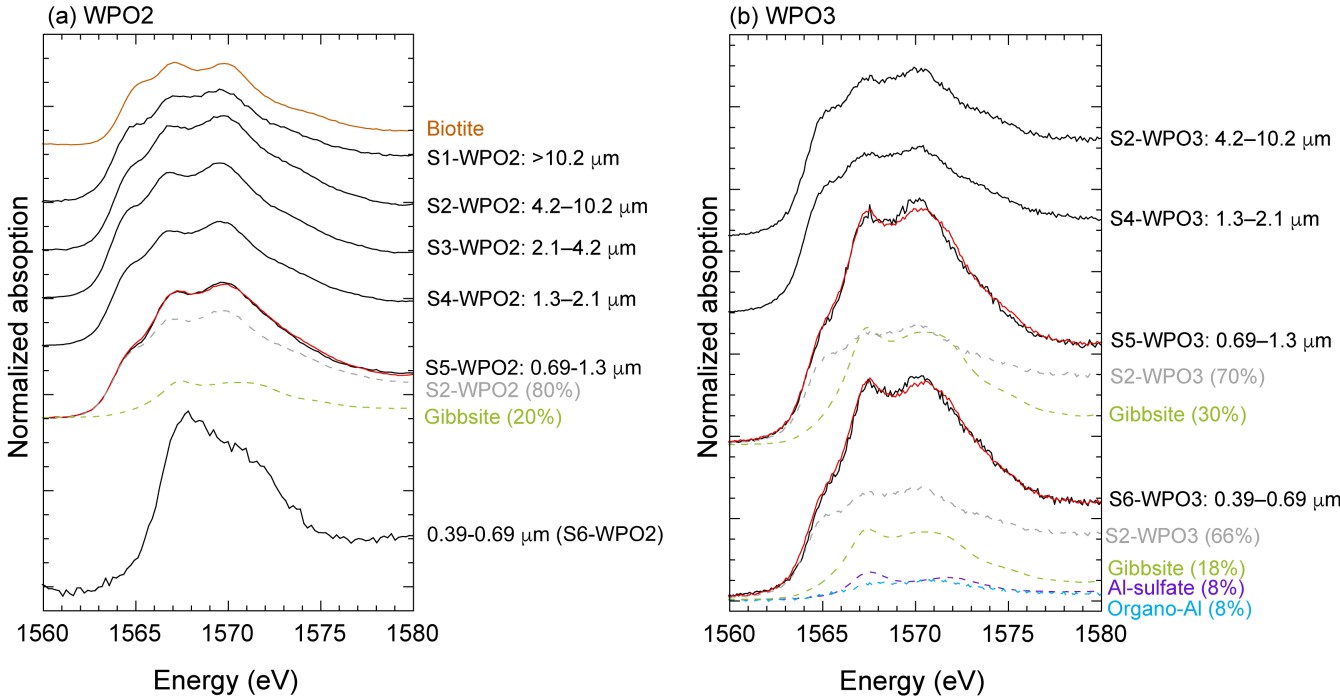

Figure 8: Al K-edge XANES spectra of (a) WPO2 and (b) WPO3. Black and red solid line showed XANES spectra for aerosol particles and fitting spectra, respectively. Colored spectra with dashed line show fitting components. The relative abundance of species identified by LCF are shown in the parentheses beside the sample name (i.e., Gibbsite (20%) for S5-WPO2).

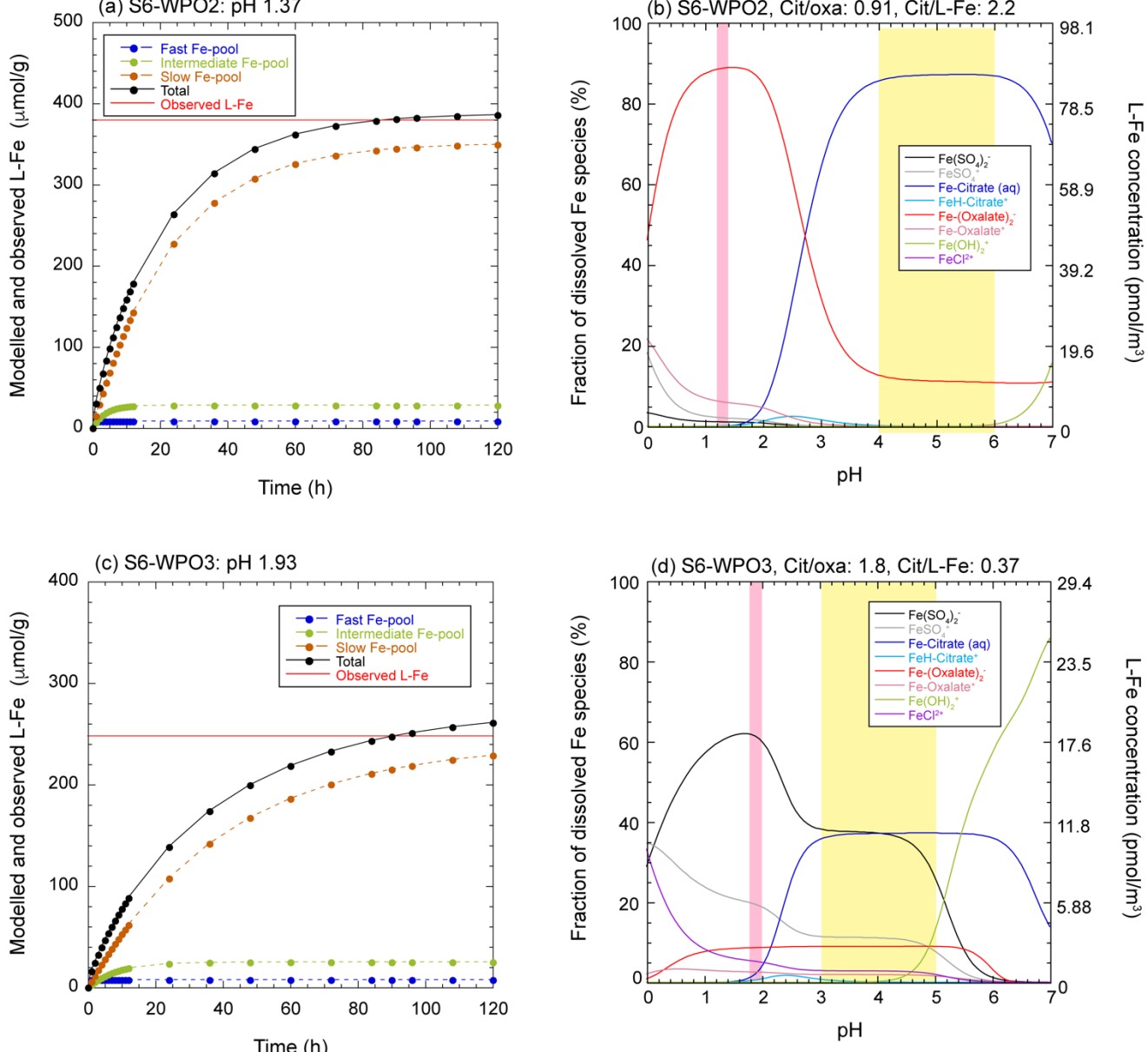

Figure 9: (a and c) dissolution curves for each Fe pool (colored dashed lines) and summation of all Fe pools (solid black line) in S6-WPO2 and S6-WPO3 as a function of dissolution time. Solid red line in these figures shows the observed L-Fe concentrations. The pH was set so that the total value reached the observed L-Fe in approximately 90 h (expected time for wet aerosol phase). (b and d) pH dependences of L-Fe species in ALW for S6-WPO2 and

700 S6-WPO3. Pink and yellow regions show the aerosol pH for the proton-promoted dissolution (same pH as in panels a and c) and stable pH regions of Fe(III)-HULIS, respectively.

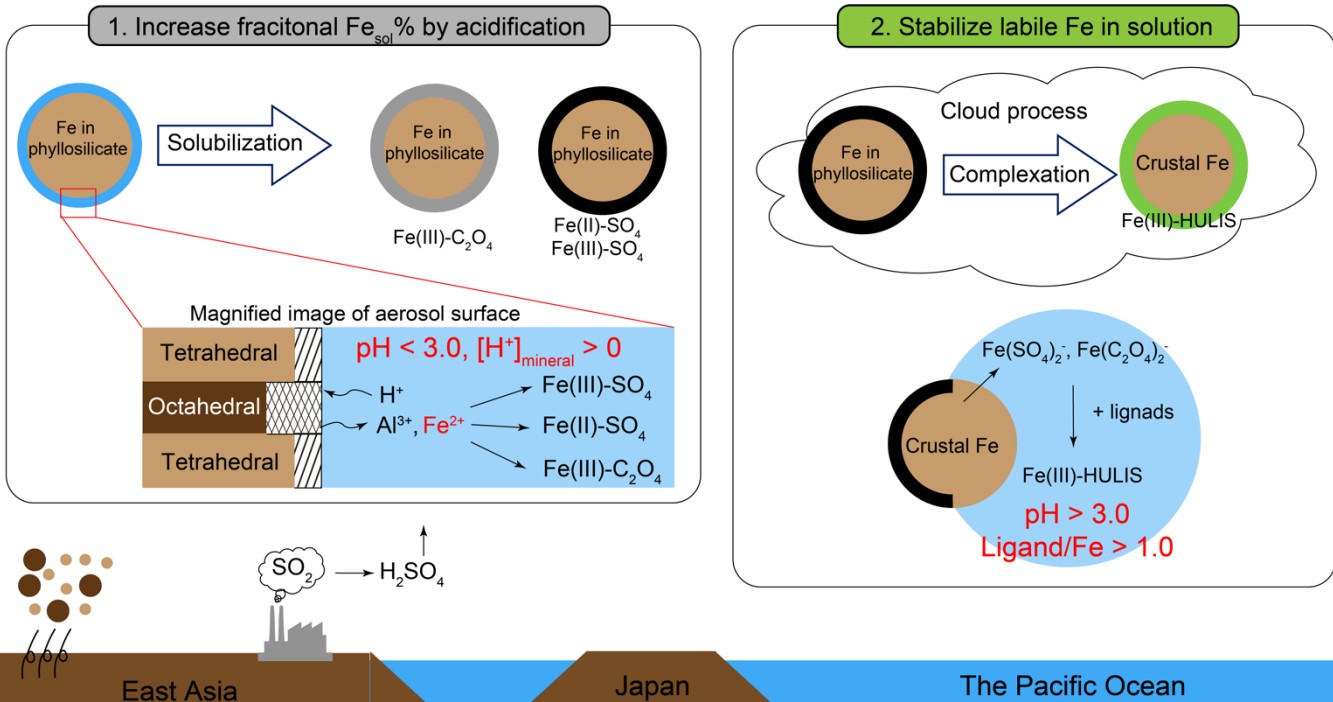

Figure 10: The schematic of alteration processes of Fe in phyllosilicate particles in $PM_{1.3}$ during transport.

Table 1. Model parameter for three Fe-pool model.

| | $pH_{PPD}$ | Expected Fe species | %Fe(0) | Dissolution rate |
|---|---|---|---|---|
| Fast | 1.0–2.0 | Ferrihydrite | Fixed at 0.9 | $\log k_{fast} = -0.50\ pH_{PPD} + 1.87$ |
| | 2.0–3.0 | Poor crystalline Fe-oxides | %FeT = –0.4 $pH_{PPD}$ +1.7 | |
| Intermediate | 1.0–2.0 | nano-size Fe-oxides | Fixed at 3.0 | $\log k_{intermediate} = -0.66\ pH_{PPD} + 0.36$ |
| | 2.0–3.0 | | %FeT = –2.0 $pH_{PPD}$ +7.0 | |
| Slow | 1.0–2.0 | Crystalline Fe-oxides | %FeT = –15.2 $pH_{PPD}$ +58.4 | $\log k'_{slow} = -0.44\ pH_{PPD} - 0.76$ |
| | 2.0–3.0 | Fe in clay mineral | | |

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
