# Peer review of "Iron (Fe) speciation in size-fractionated aerosol particles in the Pacific Ocean: The role of organic complexation of Fe with humic-like substances in controlling Fe solubility"

_Atmospheric Chemistry and Physics, 2022_

## Author Comment (AC1)

Reviewer 1

General comment

| Reviewer comments | Author reply |
|---|---|
| The processes which influence changing aerosol iron fractional solubility during atmospheric transport remain a crucial gap in our understanding. Additionally, ship-based collections of size-fractionated aerosols over the open ocean are rare. This work reports results from a field study which captures these sorts of samples and attempts to use a combination of spectroscopic and microscopic analytical methods to examine aerosol iron speciation along with chemical modeling to explore these processes. The authors find that most of the total aerosol iron was found on coarse particles of >1 μm but that iron solubility was higher on smaller particles. This observation is attributed to atmospheric processes which expose the smaller particles to environmental acidity levels beyond the inherent buffering capacity of the aerosol. Further, the authors suggest that Fe(III) complexes with humic-like substances (Fe(III)-HULIS) stabilize solubilized iron during transport prior to deposition on the ocean. This study is fundamentally sound and uses methods that are well established in the field. The novelty lies in the application of these methods on samples collected across a meridional section of the Pacific Ocean between the temperate north and south. This paper lies firmly within the scope of ACP and will be of interest to the aerosol and ocean communities following revision. The study finds that most of the total aerosol Fe is found in the coarse fraction defined as >1.3 μm but that more labile Fe was in the fine faction. Statistical tests show correlation aerosol Fe solubility and the abundance of Fe(III)-HULIS. Using models, they suggest that aerosol Fe is solubilized at pH<3 and these complexes keep the Fe stabile as pH increases as ferrihydrite precipitation is suppressed. These results are all interesting and will contribute to our collective understanding of the processing of aerosols in the atmosphere. | Thank you for your time and effort. We have carefully revised the manuscript with full consideration of the comments and suggestions provided. Please find the detailed responses below.

"Revised text as it appears in the text (in quotes, blue font)" |

| | |
|---|---|
| The text is not without weaknesses which can be improved by careful revision in some cases but perhaps not in others. Generally, the text requires careful editing for clarity and grammar. A fundamental concern of mine is that few samples were collected for this study. In total, only five marine samples and one land-based sample encompass all measurements. My concern is that results might be overinterpreted if the samples are not wholly representative. For instance, the SPO sample covers approximately 17 degrees of latitude over four days. Are there caveats to any of the conclusions because of this? Are the SPO and CPO truly different? | Long time sampling was required to determine the size-distributions of Fe concentration because the expected Fe concentration was too low to measure these factors. Moreover, the aerosol samples were collected only while the research vessel was underway to avoid contamination with ship emissions during anchorage for seawater sampling. Thus, long sampling periods with wide spatial coverage for CPO and SPO samples were required. As noted, results from observation vessels are snapshots and may be overinterpreted depending on the sample representativeness. We also regarded our samples as the average snapshot of Fe concentrations and species during each sampling. However, we believe that our observational results regarding L-Fe concentrations in marine aerosols and their relationship to Fe speciation and solubility are useful for understanding the factors determining $Fe_{sol\%}$ in marine aerosol particles because of the limited observational data on the size dependence of these factors. Therefore, we think that the results provide rare and important information.

"This was a case study on the relationship between $Fe_{sol}$% and Fe species in size-fractionated aerosol particles collected from the Pacific Ocean." |
| Some of the conclusions hinge on inferences from the SXTM single particle analysis. More generally on this point, the comparisons of the five different samples was given far less emphasis than the geochemical modeling which I think is the reverse of the proper strategy. | We have added descriptions about the comparison of the internal mixing state of mineral dust with Na and OCs reported in previous studies. The internal mixing particles between mineral dust and SSA (dust-SSA) were found in the WPO and other oceanic regions. It is considered that dust-SSA particles were formed through in-cloud process. Moreover, a previous study revealed similarities in OC species in SSA and dust-SSA. As (i) SSA is ubiquitously present in the marine atmosphere and (ii) chemical compositions of marine cloud waters are influenced by SSA, the internal mixing between mineral dust and SSA through the cloud process contributes to Fe(III)-HULIS formation. These results play a significant role in the discussion on the formation of Fe(III)-HULIS with increasing pH of mineral dust.

"Similar internal mixing particles between mineral dust and SSA have been found not only in the Pacific Ocean, but also in other regions (Okada et al., 1990; Niimura et al., 1998; Wagner et al., 2008; Kandler et al., 2017; Adachi et al., 2020; Kwak et al., 2022; Knopf et |

<table>
<tr>
<td></td>
<td>al., 2022). It is considered that the internal mixing of mineral dust and sea salt is formed by cloud processes (Niimura et al., 1998; Formenti et al., 2011). A recent study found internal mixing particles between aged sea salt, mineral dust, S, and OCs in submicron aerosols collected from the North Atlantic Ocean, of which OCs species were similar to those in SSA (Knopf et al., 2022). This result is similar to the microscopic observation results (Figs. S7 and S8). Since (i) SSA is ubiquitously present in the marine atmosphere and (ii) the chemical composition of marine cloud water is influenced by SSA, the internal mixing of mineral dust with SSA in cloud water may play a significant role in the organic complexation of L-Fe in aerosol particles in the marine atmosphere."</td>
</tr>
<tr>
<td>The authors rely to heavily on references to the supplemental material. For example, on page 11 the authors make eight references to figures and tables in the supplement. This is too much. I encourage a careful consideration of what data and figures are required and these materials be included in the main paper. As an example, I am surprised that Table S1 which includes crucial information about the aerosol size fractionation scheme, filter type, and sample description scheme is relegated to the supplement. Indeed, Equation 3 in the Supplement defines the term [H+]mineral which is used throughout the paper. This equation needs to be in the primary article.</td>
<td>Thank you for your comment. We have reconstructed the contents of figures and tables in the manuscript and the supporting information. We have added descriptions of aerosol size for a cascade impactor and explanations about $[H^+]_{mineral}$ in the main text (please see section 2.4).</td>
</tr>
</table>

Specific comments

| No. | Reviewer comments | Author reply |
|-----|-------------------|--------------|
| 1.1 | Line 76: "...decreases a saturation index of Fe…" Revise for clarity | We have improved the sentence in the following manner: "Moreover, the formation of organic complexes with L-Fe in the aqueous phase promoted further Fe dissolution from the aerosol particles to aerosol liquid water (ALW)." |

| | | |
|---|---|---|
| 1.2 | Line 114-115: I don't understand why the label PM$_{1.3-10.2}$ is used for coarse particles but PM$_{1.3}$ is used for fine. Both represent a range of size classes but only the former indicates as such. PMcoarse and PMfine would be appropriate or, if the desire is to maintain the current form, perhaps PM$_{1.3-10.2}$ and PM$_{<1.3}$. | Thank you for your comment. In general, aerosol particles finer than 2.5 μm are described as PM$_{2.5}$. The description is available if the cut-off diameter is other than 2.5 μm. Therefore, we described fine aerosol particles as PM$_{1.3}$. In contrast, coarse aerosol particles cannot be described as PM$_{10.2}$ because PM$_{10.2}$ encompasses PM$_{1.3}$. We have rephrased PM$_{1.3-10.2}$ as PM$_{>1.3}$. It should be noted that PM$_{1.0-2.5}$ in section 4.2 has not been rephrased because we have focused on aerosol particles ranging from 1.0 to 2.5 μm. |
| 1.3 | How were blank concentrations in ng/cm2 converted to pg/m3? | We have added the method used for unit conversion of the filter blank concentrations of Fe and Al. "The blanks of Al and Fe in the PTFE filter were 0.306±0.352 and 0.335±0.340 ng/cm$^2$, respectively. The unit of filter blank concentrations were converted from ng/cm$^2$ to ng/m$^3$ by the following equation: $$Filter\ blank\ (ng/m^3) = \frac{filter\ blank\ (ng/cm^2) \times filter\ area\ (cm^2)}{Total\ flow\ for\ each\ sampling\ (m^3)} \qquad (Eq.\ 1)$$ As a result, the filter blank concentrations of Al and Fe were a few pg/m$^3$." |
| 1.4 | Line 136: How long were the aerosol samples stored prior to labile metal extraction? Were the samples frozen? | Aerosol samples were stored in a dry desiccator at 20% RH and room temperature (approximately 20 ℃). |
| 1.5 | Line 148: More details are needed on the ion analysis. What were the levels of detection? Were there issues with the stability of ammonium and/or oxalate? | We have added descriptions for LOD and stability and ammonium, nitrate, and oxalate. In the case of nitrate in coarse aerosols, nitrate was mainly present as NaNO$_3$ because almost no NH$_4^+$ was detectable in the fraction. In contrast, some NH$_4$NO$_3$ may evaporate in PM$_{1.3}$. According to previous studies, during sampling, negative artifacts have negligible effects on ammonium sulfate and oxalate concentrations (Yao et al., 2002; Bian et al., 2014). "The detection limits of the ICS-1000 for Na$^+$, NH$_4^+$, K$^+$, Mg$^{2+}$, Ca$^{2+}$, Cl$^-$, NO$_3^-$, SO$_4^{2-}$, and C$_2$O$_4^{2-}$ were 0.556, 0.464, 1.15, 0.726, 1.50, 5.62, 15.0, 18.8, and 33.2 ng/mL, respectively. Among the targeted ions, the lowest and highest filter blank concentrations were 0.0687 and 32.4 ng/cm$^2$ for Mg$^{2+}$ and SO$_4^{2-}$, respectively (Sakata et al., 2018). |

| | | After the unit conversion of the filter blank from ng/cm$_2$ to ng/m$_3$ using Equation 1, the highest filter blank concentration was 4.47 ng/m$_3$ SO$_4^{2-}$. Semi-volatile compounds (e.g., NH$_4$NO$_3$) were affected by negative artifacts during sampling. The negative artifact effect was unlikely to be significant because most nitrates were present in PM$_{>1.3}$ with a small concentration of NH$_4^+$. However, some NH$_4$NO$_3$ present in PM$_{1.3}$ may be affected by the negative artifact. The negative artifacts of oxalate and ammonium sulfate are usually negligible in IC analyses (Yao et al., 2002; Bian et al., 2014)." |
|---|---|---|
| 1.6 | Line 156: [H+]mineral must be defined in the main article. | Thank you for your comment. We have moved the explanation about [H$^+$]$_{mineral}$ from the Supplemental Information to the manuscript (please see section 2.4 in the manuscript). |
| 1.7 | Line 171: What is LCF? | We apologize for not providing an explanation for LCF, which is an abbreviation for linear combination fitting. We have spelled out LCF in the first appearance in the manuscript. |
| 1.8 | Equations 8 and 9: What do the constants 1.76 and 0.76 represent? | The intercept means the dissolution rate (log k) at pH 0, which is estimated by extrapolation from the experimental results (pH 1, 2, and 3) of the previous study (Shi et al. 2011). It should be noted that the equation is applicable between pH 1–3 in this study. |
| 1.9 | Line 219: What is pH (optional)? | We apologize for the ambiguity of the sentence. We intended to state that pH is variable in the equation described in Table 1. We have improved the sentence as follows: "Finally, the dissolution curves at various pH values are described in Table 1. This curve with the pH of each sample was used to explain the observed L-Fe within the expected transport time." |
| 1.10 | Line 299: WPO1<WPO2<WPO3< SPO≈CPO Does this ranking indicated aerosol abundance, particle diameter, distance, something else? | The order shows the distance of the transported distance from the expected source region. The description was modified as below: "The relative abundance of ferrihydrite increased with decreasing aerosol diameter and increasing transportation distance (Fig. 7a, transport distance: WPO1 < WPO2 < WPO3 < SPO ≅ CPO)." |
| 1.11 | Line 349: What is ODc-pre? | We apologize for this mistake. We have removed it. |

| | | |
|---|---|---|
| 1.12 | Line 402: The text states that the pH of cloud water decreases by 0.03 units but the next sentence states that there is an increase in aerosol pH by cloud processes. Please clarify. | We apologize for the ambiguity of the sentence. We have modified the sentence as follows:

"Given that 500 nm of phyllosilicate particles with a 100 nm-thick water layer at pH 1.0 was incorporated into typical marine cloud water (diameter 10 μm, pH 4.0, Boris et al., 2016; Kim et al., 2019; Shah et al., 2020), the pH of aerosol particles was 3.97." |
| 1.13 | Line 435: I do not think that changing PM1.3-2.3 to PM1.0-2.5 is simplifying. | We removed the term $PM_{1.3-2.3}$ from the manuscript, and the sentence has been improved as follows:

"In contrast, aerosol particles in stage-4 ($PM_{1.0-2.5}$) did not have a high $Fe_{sol}$% because mineral dust in the fraction was not acidified because of the negative $[H^+]_{mineral}$." |
| 1.14 | Figure 1: Suggest removing the gray arrows as these do not represent samples included in this paper. | Thank you for your comment. We have removed the gray arrows from the figure. |
| 1.15 | Figure 2: Air mass back trajectories could be moved to the supplement. | According to your suggestion, the figure has been moved to the Supplementary Information. |
| 1.16 | Figure 6: I am not sure that the panes b, c, and e are necessary for the main paper and could be moved to the supplement as many of these images already are. 6(e) is not described in the caption. | We considered that the XANES spectra of several samples must be shown in the manuscript because spectra information is the most important data of spectroscopic studies. We are sorry for the lack of explanation about Fig. 6(e). We have included the explanation for Fig. 6(e). |
| 1.17 | Figure 7: Much more detail is required in the caption. Clarify that S(n) refers to specific size classes. What are the significance of Particle 1, Particle 2, and Particle 3? What is SRFA? | The Al K-edge XANES spectra of single particles 1 to 3 and SRFA (organo-Al) have been removed from Figure 7. We have also removed Figure 8 from the manuscript. We apologize for the ambiguous figure captions. The aerosol diameter of each sample has been described in the figure rather than in the caption. We have improved the caption for Figure 7 as follows:

"Figure 7: (a) Fraction of Fe species in each sample determined by Fe K-edge XANES spectroscopy. Iron K-edge XANES spectra of size-fractionated aerosol particles collected in (b) WPO2 and (c) NOTOGRO. (d) a scatter plot between fraction of |

| | | Fe(III)-HULIS and $Fe_{sol}\%$ in $PM_{1.3}$." |
|---|---|---|
| 1.18 | Figure 8 is not necessary. | According to your suggestion, we have removed the figure from the manuscript. |

Reviewer 2

General comments

| Reviewer comment | Author reply |
|---|---|
| Organic ligands have been postulated to enhance aerosol iron solubility, but the chemical speciation of Fe complexes in size-resolved aerosols is not characterized well. The authors analyzed Fe species in size-fractionated aerosol particles over the Pacific. The X-ray spectrum analysis using reference materials indicated that fine particles contained ferric organic complexes with humic-like substances. The Fe(III)-HULIS was suggested to be formed during transport to the Pacific. The results presented in this paper contribute to better understanding of Fe cycles. I have some comments and questions to improve this paper. | Thank you for the time and effort required to review our manuscript. We have carefully revised the manuscript with full consideration of the comments and suggestions provided. Please find the detailed responses below.

"Revised text as it appears in the text (in quotes, blue font)" |

Specific Comments

| No. | Reviewer comments | Author reply |
|---|---|---|
| 2.1 | General reply to the reviewer (Dr. Ito) in particular about the contribution of the anthropogenic Fe. | The importance of anthropogenic Fe and the suppression effects of dust/liquid ratio on Fe dissolution were emphasized in several comments given by Dr. Ito. We think that elemental ratios and statistical analysis added in the revise version showed the importance of mineral dust rather than anthropogenic emission as a source of Fe in aerosol samples in this study (please see our replies to Comments 2.2, 2.15, and 2.16). The reviewer also suggested the significance of anthropogenic Fe due to the higher $Fe_{sol}$% compared with laboratory experiments. However, aerosol diameter with high $Fe_{sol}$% (<1.3 µm) was different from those for general laboratory experiments (TSP or PM10). The $Fe_{sol}$% of our TSP samples was up to 7.69%. This value was within an upper limit suggested by you (< 15% at pH 1.0 and for 120 h). Moreover, $Fe_{sol}$% higher than 15% has been reported in laboratory experiments on Fe dissolution from mineral particles (Shi et al., 2011). Therefore, we do not consider that the emission source of L-Fe should not be estimated based only on the $Fe_{sol}$% (Please see our reply to Comment 2.22). |

| | | | |
|---|---|---|---|
| | | | As stated by Dr. Ito, Fe dissolution from mineral dust is suppressed at higher dust/liquid ratios. Since the dissolution model was constructed using experimental data at a dust/liquid ratio of 60 mg/L, the model may overestimate L-Fe concentration. However, lower pH condition is needed to explain the observed L-Fe with consideration of the suppression effects. Therefore, it does not significantly affect our conclusion that aerosol acidification is the dominant process for solubilizing Fe in aerosol particles (Please see our reply to the comment 2.10). |
| 2.2 | P2., l.30: Even though EF of Fe is close to one, L-Fe in fine particles can be derived from anthropogenic sources due to much higher solubility, as is indicated by Fe stable isotope ratios and the model estimates over the northwestern Pacific. This lower Fe solubility for mineral dust is partly because the high dust/liquid ratio due to low water content in mineral dust could suppress the Fe dissolution even in acidic condition over polluted regions, in addition to the buffering capacity of calcite. Moreover, the Fe dissolution rate for mineral dust is much slower than fly ash. Please see below and consider rephrasing L-Fe for mineral dust in PM1.3 throughout the paper. | | According to your suggestion, L-Fe from mineral dust was rephrased as L-Fe in fine aerosol particles as the existence of anthropogenic Fe cannot be completely ruled out. However, we still considered that anthropogenic Fe was not the dominant L-Fe source in our samples.

Anthropogenic Fe with a negative Fe isotope ratio ($\delta^{56}$Fe) is emitted as Fe-oxides with a small amount of coexisted elements, including Al. The relationship between EF and $\delta^{56}$Fe needs to be further evaluated, but the presence of Fe-oxides should increase EF of Fe. The EF of Fe in $PM_{1.3}$ collected near the source regions was approximately 2.0 or more. Therefore, the influence of anthropogenic Fe-oxides is unlikely to be significant when the EF of Fe is approximately 1.

In the case of fly ash, it is difficult to distinguish between mineral dust and coal fly ash because of the similar EF between Fe and $\delta^{56}$Fe (ash: approximately –0.1%). However, coal and coal fly ashes enrich cobalt (EF ~ 10) compared to mineraldust. If coal fly ashes are the dominant L-Fe source of $PM_{1.3}$, the expected EF of our samples is 3.0 or more. However, EF of Co in $PM_{1.3}$ was almost 1.0, indicating lower amount of coal fly ash in the sample. Furthermore, the correlation between $Fe_{sol}$% and EF of Pb was likely a pseudo-correlation, with little direct relationship between these components due to the small partial correlation factor (–0.15). Thus, the observed data suggest that the effects of anthropogenic Fe on $Fe_{sol}$% are not significant. Detailed discussions are described in |

| | | | |
|---|---|---|---|
| | | section 3.2 in the revised manuscript. Information regarding the dust/liquid ratio has been detailed in 2.10. |
| 2.3 | p.3, l.72 Please define the average pH. | We apologize for the ambiguous description. We have improved the sentence as follows: "Based on the $Fe_{sol}$% and speciation results, the expected pH required for L-Fe concentration in the aerosol samples by proton-promoted dissolution within the transport time ($pH_{PPD}$) was evaluated using a conceptual model following first-order iron dissolution." |
| 2.4 | p.4, l.81: Please specify previous studies for HULIS on mineral dust or other types. How does it form on mineral dust or other types? | We have added descriptions of the source and formation processes of atmospheric HULIS. "Atmospheric HULIS in marine aerosols are formed by atmospheric processes and direct emissions from the ocean surface (Deng et al., 2014; Chen et al., 2016; Santander et al., 2021), whereas soil-derived organic matter is generally not an important source of atmospheric HULIS (Graber and Rudich, 2006; Spranger et al., 2020)." |
| 2.5 | p.4, l.83: Please specify previous studies for siderophore on mineral dust or other types. How does it form on mineral dust or other types? | There are limited data regarding the emission source and siderophore formation in aerosol particles, but biological activities on mineral dust and in cloud water are considered as the formation processes in the aerosol particles. "In addition, siderophores have been detected in aerosols, rainwater, and cloud water, which are likely formed by biological activities in mineral dust and cloud water (Cheize et al., 2012; Sullivan et al., 2012; Vinatier et al., 2016)." |
| 2.6 | p.6, l.137: Please specify the filter. Sub-micron particles for Stage 6 samples might penetrate the filter regardless its chemical form including nano particulate form. How do you consider this? | Insoluble particles in the extracted solution were filtrated using a hydrophilic PTFE filter ($\phi$: 0.20 μm). The aerodynamic diameter for stage 6 ranged from 0.39 to 0.69 μm, while smaller particles were collected in stage 7. Therefore, there are almost no nanoparticles passing through the PTFE filter. We have specified the filter type in the manuscript. "The extracted solutions were acidified to 0.15 mol/L after filtration of insoluble particles using a hydrophilic syringe PTFE filter (□:0.20 □m, Dismic®, 25HP020AN, Advantec, Japan)." |

| | | |
|---|---|---|
| 2.7 | p.6, l.157: How do you consider the external mixing of Fe-bearing particles with the main component of the marine aerosols mentioned in introduction? | We have added the detailed method for calculation the available protons in mineral dust ($[H^+]_{mineral}$) in section 2.4. We calculated $[H^+]_{mineral}$ with the assumption that Fe-bearing particles externally mixed with sea spray aerosol, $(NH_4)_2SO_4$, and $NH_4NO_3$. $[H^+]_{mineral}$ refers to the maximum amount of $H_2SO_4$ and $HNO_3$ that can be internally mixed with iron-bearing particles. |
| 2.8 | Figure 6: Fig. 6 appears before Fig. 2. Please also correct the caption (d) and check the consistency of Fe species with (a). | Thank you for pointing this out. The XANES and EXAFS spectra of key Fe species are shown in Fig. 2, and Figure 6b has been removed. In addition, we have corrected the legend of Fig. 6d (Fig. 6c in the revised version). |
| 2.9 | p.7, l.176: Please also describe the distinction of Fe(III)-HULIS from ferrihydrite and goethite. | We have added the following explanations about spectrum differences between Fe(III)-HULIS, ferrihydrite, and goethite.

 "These species were distinguished from Fe(III)-HULIS because Fe(III)-HULIS has a flat peak at 7125–7135 eV (Fig. 2). In the case of ferrihydrite and goethite, these XANES spectra have a flatter peak than hematite, but the width of the peak is narrower than that of Fe(III)-HULIS (Fig. 2). Furthermore, the EXAFS spectrum of Fe(III)-HULIS was clearly different from that of ferrihydrite, goethite, and hematite. Fe(III)-HULIS has a single peak at 7–9 Å in the k-space, whereas Fe-(hydr)oxides have two peaks in the same region. Based on these spectral differences, the Fe species in the aerosol particles were determined using the LCF method." |
| 2.10 | p.8, l.202: These equations are not valid at the high dust/liquid ratio due to low water content in mineral dust, which could suppress the Fe dissolution even in acidic solution. How do you consider the degree of the suppression? | We agree that dissolution of Fe from mineral dust is suppressed by the high dust/liquid ratio. The calculation may overestimate modeled L-Fe concentration because it does not take into account the suppressive effect of high dust/liquid ratios on Fe dissolution. A pH lower (or higher aerosol acidity) than $pH_{PPD}$ would be required to account for the observed L-Fe concentration with consideration of the suppression effect. Therefore, it does not significantly affect the conclusion that aerosol acidification is important for explaining high L-Fe concentrations. According to your comment, we have added descriptions regarding the overestimation of L-Fe concentration due to the suppressive effects in the |

| | | | |
|---|---|---|---|

manuscript.

"It should be noted that these kinetic parameters are estimated using the experimental data with a solid/liquid ratio of 60 mg/L. The actual aerosol dust/liquid ratios are predicted to be as high as 3000 g/L, which may suppress the dissolution of Fe from the aerosol particles (Shi et al., 2011). Our calculation results may have overestimated the modeled L-Fe concentration at $pH_{PPD}$ with a high dust/liquid ratio. In other words, a lower pH (or higher aerosol acidity) than the predicted $pH_{PPD}$ is required to account for the observed L-Fe concentration, while considering the suppression effect. Therefore, $pH_{PPD}$ can be recognized as the upper pH limit to explain the observed L-Fe concentration by proton-promoted dissolution."
* * *
**2.11** — p.8, l.210 and l.211: The cloud cycle and dissolution time depend on the transport pathway of the particles. Please specify the method and references to justify the 12 hours and half of the transportation time.

We apologize for the absence of the reference about cloud cycles. As noted, the time of the cloud process would depend on the transport path, but it is not easy to predict the exact time. We reconsidered the time of the cloud cycle based on the global average residence time of aerosol particles and clouds (Pruppacher and Jaenicke, 1995). As a result, pH was estimated by assuming that aerosol particles were in the evaporated state during 75% of the transport time. In this calculation, $pH_{PPD}$ is slightly higher than the estimated pH in the previous version.

"Mineral dust is expected to undergo several condensation-evaporation cycles during transport (Pruppacher and Jaenicke, 1995). Proton-promoted Fe dissolution occurred during the evaporation state (wet aerosol), whereas aerosol particles were taken in cloud water during the condensation phase. According to a previous study, the global average residence times for aerosol particles before uptake by clouds and within the cloud in an air parcel are up to 12 h and 3 h, respectively (Pruppacher and Jaenicke, 1995). Based on these residence times, aerosol particles are expected to exist in an evaporative state (wet aerosol) for approximately 67–80% of their

| | | |
|---|---|---|
| | | transport time. In this study, the estimation of pH was estimated assuming that aerosol particles spent the evaporated state in 75% of transport time (approximately 90 h for the WPO and 130 h for CPO and SPO)." |
| 2.12 | p.8, l.214: The dissolution curve is fitted to the mass concentration. Please specify the dissolution rates of biotite and illite after normalizing the dissolution rates to the mineral mass. | Labile Fe concentration in aerosol particles was normalized by mineral mass concentration. Mineral dust in the atmosphere was estimated as follows: total Al concentration divided by the average abundance of Al in the continental crust. |
| 2.13 | p.8, l.225: How do you consider calcite? How do you also consider the external mixing of Fe-bearing particles with the main component of the marine aerosols mentioned in introduction? | We did not consider the effect of calcite in E-AIM calculation due to the following reasons: (i) mineral dust in fine aerosol particles was likely acidified beyond the buffer capacity of calcite, (ii) calcite was not the dominant Ca species in fine aerosol particles collected in Japan, even though aerosol samples were collected in the dust season (Miyamoto et al., 2020), and (iii) Ca speciation indicated that calcite was not the dominant Ca species in the S6-WPO2 and S6-WPO3 samples (please see below figure, for only review). Moreover, we used E-AIM for the estimation of ALW rather than pH.

"In this study, the buffering effect of calcite in the equilibrium calculation was not considered because (i) mineral dust was likely acidified beyond the buffering capacity of calcite, and (ii) calcite in fine aerosol particles was altered to $CaSO_4 \cdot 2H_2O$ and $CaC_2O_4$ during transport from the source region of Asian dust to Japan (Takahashi et al., 2008; Miyamoto et al., 2020)." |
| 2.14 | p.9, l.251: Please show the enrichment factors for Beijing aerosol and NOTOGRO. | Enrichment factors and fractional Fe solubilities in NOTGORO sample are shown in Fig. |

| | | |
|---|---|---|
| | p.10, l.284: Please show Fe solubilities for Beijing aerosol and NOTOGRO. | S1. Enrichment factors of NOTOGRO samples were also approximately 1.0 regardless of aerosol diameter. The $Fe_{sol}$% in coarse fraction was approximately 1.0%, whereas that in stage-5 and stage 6 was 19.7% and 34.4%, respectively. In the case of Beijing dust, EF and $Fe_{sol}$% were 0.85 and 0.53%, respectively. |
| 2.15 | p.10, l.276: How do you reconcile non-anthropogenic Fe source with the positive correlation between Fe solubility and EF of Pb?

p.10, l.280: How do you reconcile the positive correlation between Fe solubility and non-sea-salt sulfate with non-anthropogenic Fe source? | First, we have added Fig. 5 regarding the correlation between $Fe_{sol}$% and trace elements and nss-$SO_4^{2-}$ instead of Table S3. As described in 2.1, the correlation between $Fe_{sol}$%, Pb, and Cd does not always guarantee the significant effect of anthropogenic Fe on $Fe_{sol}$% because mineral dust and anthropogenic aerosol and gases were externally mixed during transport.

A correlation between $Fe_{sol}$% and nss-$SO_4^{2-}$ has been reported when the Fe in mineral dust was solubilized by coal-derived $SO_2$ (Wong et al., 2020). In fact, there is a significant correlation between $(NH_4)_2SO_4$ and sulfate-containing dust in Asian dust (Sullivan et al., 2007). Therefore, a positive correlation between L-Fe, Pb, Cd, and nss-$SO_4^{2-}$ can be observed, even if these components are not emitted from the same emission source. |
| 2.16 | p.10, l.290: How do you reconcile this with the positive correlation between Cd or Pb and non-sea-salt sulfate? How do you explain low non-sea-salt sulfate concentrations over SPO? Please show non-sea-salt sulfate concentration which can be attributed to biogenic S emission alone. | The positive correlation between $Fe_{sol}$% and Pb, Cd, and nss-$SO_4^{2-}$ has been described in 2.15.

The low concentration of nss-$SO_4^{2-}$ in SPO could be attributed to the low influence of anthropogenic emissions. A recent observational study on S isotope in $PM_{1.0}$ collected from the coast of New Zealand showed that almost all nss-$SO_4^{2-}$ in $PM_{1.0}$ had biogenic origins. The S isotope in submicron aerosol showed biogenic origins (Calhoun et al., 1991). For these reasons, it is reasonable that most of the nss-$SO_4^{2-}$ in CPO and SPO is of biogenic origins (Please see the last paragraph in section 3.2). |
| 2.17 | p.10, l.287: How do you confirm this before the 7-day backward trajectories? | The 10-day backward trajectories for SPO and CPO show little or no terrestrial influence. |
| 2.18 | p.11, l.315: Please show biotite fraction for Beijing dust quantitatively. | We have added the relative amounts of Fe species in Beijing dust. The relative biotite abundance was 36%.

"Beijing dust also contained Fe(II)-sulfate and Fe(III)-sulfate with ferrihydrite and |

| | | |
|---|---|---|
| | | biotite. Relative abundances of these species to total Fe were 9 %, 11 %, 44 %, and 36 %, respectively (Fig. S4d)." |
| 2.19 | p.11, l.319: The biotite fraction in S6-WPO2 is higher than that in S6-NOTOGRO. The biotite fraction in the S5-WPO3 is also higher than that S5-NOTOGRO. These results rather suggest that biotite in fine particles is relatively insoluble. How do you explain higher biotite fraction over the oceans than NOTOGRO? | The sampling years for size-fractionated aerosol particles are different between the Western Pacific and NOTOGRO samples. Therefore, unfortunately, the quantitative comparison of Fe species between these samples was difficult. However, we still consider that the NOTOGRO sample can be used as an analog for Fe species in the $PM_{1.3}$ altered during the transport from East Asia to Japan (before reaching the West Pacific). |
| 2.20 | p.11, l.335: Please show the fraction of Al species quantitatively. | We have added the abundance of Al species in the manuscript. "Gibbsite was found in S5-WPO2 and S5-WPO3, with which abundances of 20 % and 30 % in total Al, respectively (Figs. 8a and 8b). The S6-WPO3 contained Al-sulfate and organic complexes of Al (organo-Al), gibbsite, and phyllosilicates, and their abundances were 8 %, 8 %, 18 %, and 66 %, respectively. The presence of organo-Al and Al-sulfate in S6-WPO3 is consistent with that of Fe(III)-sulfate in this sample (Figs. 7a and 8b)." |
| 2.21 | p.12, l.360: Non-spherical dust particles can be converted to spherical particles when they are intensely altered. Thus, the irregular shapes rather suggest that phyllosilicate particles in S6-WPO2 are relatively unaltered. How do you reconcile this intensely altered particles with irregular shapes? | We are unsure of the effect of weathering on the conversion of non-spherical dust particles into spherical particles. Microscopy-based images of mineral dust before and after proton-promoted dissolution in laboratory experiments have been published. However, mineral particles are irregularly shaped even after undergoing acidification at pH 1.0 (Wang et al., 2018; Xie et al., 2021). Therefore, it is difficult to determine the degree of acidification based on the change in the shape of mineral particles. Wang et al. (2018): https://doi.org/10.1016/j.jes.2018.04.012 Xie et al. (2021): https://doi.org/10.1016/j.atmosenv.2021.118436 |
| 2.22 | p.13, l.375 and Fig. 9: Why don't you show the dissolution curve from the Beijing dust? In the laboratory experiments, Fe solubility for mineral dust has not reached more than 15% at pH 1 for the proton-promoted dissolution time up to 120 (h), in contrast to fly ash. Moreover, Eq. 7 is not valid at the high dust/liquid ratio due to low water content in | Fe dissolution experiments have been previous performed in the particle size range close to $PM_{10}$ or TSP. Therefore, the results of laboratory experiments should be compared to the observed $Fe_{sol}$% of TSP rather than $PM_{1.3}$. The $Fe_{sol}$% of TSP ranged from 0.967% to 7.69%, which were lower than the upper limit of $Fe_{sol}$% as you have mentioned. Therefore, |

mineral dust, which could suppress the Fe dissolution even in acidic condition over polluted regions. Thus, it is extremely hard to accept such high Fe solubility for the samples with no evidence from the laboratory experiments and field observations for mineral dust near the source regions. The results presented in this paper rather suggest that L-Fe in fine particle is mainly derived from anthropogenic source.

it is difficult to directly compare the $Fe_{sol}$% of our $PM_{1.3}$ samples with the results of the laboratory experiments.

We are unsure why the upper limit of $Fe_{sol}$% for mineral dissolution (pH 1.0, 120 h) is 15%. Indeed, the dissolution rate of the slow Fe pool using this study was modified based on the XAFS results as the dissolution rate of biotite is approximately an order of magnitude higher than that of illite. As a result, our model can explain observed L-Fe concentration within a shorter dissolution time compared to the original model (Shi et al., 2011). However, even if $Fe_{sol}$% was calculated using original data, approximately 27% of the Fe in Beijing dust was solubilized at pH 1 for 120 h. Therefore, $Fe_{sol}$% can exceed 15% even for mineral particles.

Finally, the advantage of this study is that it is possible to distinguish between acidified and non-acidified aerosol. This advantage facilitates comparison with $Fe_{sol}$% determined by the laboratory experiments (all particles were acidified). However, as Beijing dust (CRM No. 28) is a TSP sample, it contains acidified and non-acidified particles. Therefore, it is difficult to estimate $pH_{PPD}$ using L-Fe data obtained using MQ extraction.

| 2.23 | p.14, l.412: How do you reconcile this with the acidifications of mineral dust by sulfate derived from biogenic S mentioned in this paper? | Longo et al. (2016) showed increasing $Fe_{sol}$% in Saharan dust in response to aerosol acidification during transport in the Atlantic Ocean, even if the dust particles did not pass over the polluted region. Therefore, aerosol acidification may be promoted in the marine atmosphere. As mentioned in 2.15, the S source in CPO was mainly derived from biogenic S, but further studies are needed to study the effect of biogenic S on $Fe_{sol}$%. |

"By contrast, the CPO sample did not pass over the polluted region (Fig. S1b), and positive $[H^+]_{mineral}$ and low $pH_{PPD}$ were observed in S6-CPO (Figs. 6a and S12a). Aluminosilicate particles react with sulfate through cloud processes, even if the particles do not pass over the polluted region (Fitzgerald et al., 2015). Moreover, a previous study reported that the $Fe_{sol}$% in Saharan dust was increased by aerosol acidification by nss-$SO_4^{2-}$ during long-range transport in the Atlantic Ocean (Longo

| | | et al., 2016). Therefore, similar reaction processes can promote the acidification of the CPO sample. Although nss-$SO_4^{2-}$ in the pelagic region is thought to be derived from biogenic origins (Calhoun et al., 1991; Li et al., 2018), further studies are required for determining the effect of biogenic S on the increase in $Fe_{sol}$%." |
|---|---|---|
| $Fe_{sol}$ | | |

---

## Author Response (AR2)

| 1. | The paper has been improved substantially. I have some comments for a clarification of my concern. | Thank you for reconsideration of our manuscript. We have carefully revised the manuscript with full consideration of the comments and suggestions provided. Please find the detailed responses below. |
|---|---|---|
| 2. | l.26, l.320, l.342: The sentences associated with the enrichment factor of Fe should be revised, according to l.350. | After careful considerations by all the co-authors, we would like to leave these sentences as they were at this time, for the detailed reasons as described below. L26: As for the abstract, we think it would be good if emission sources of the Fe emissions (mineral or anthropogenic) in the analyzed aerosol are clear. Therefore, there is no need to talk about Fe of anthropogenic origin, which did not have a strong influence on our samples. L320: We mentioned in section 3.1 that Fe in TSP samples was derived from mineral dust due to the unity of EF. However, it is within a range of regular usage of EFs to infer that Fe in aerosol particles is derived from mineral dust when the EF of Fe is 1. Since about 80% of Fe in TSP was in coarse fraction with a minor contribution of anthropogenic Fe, Fe in TSP was derived from mineral dust. Furthermore, we mentioned the correlation between $Fe_{sol}$% and EF of Pb in section 3.2. Therefore, we are puzzled by the mention of anthropogenic Fe in section 3.1. L342: The sentence mentions anthropogenic Fe oxides with minor coexistence elements (expected EF > 1) rather than coal fly ash. Therefore, we do not revise the sentence. |
| 3. | l.334: Please correct to $PM_{1.3}$. | Thank you for pointing this out. We have corrected it. |
| 4. | l.360: The good correlations of nss-$SO_4^{2-}$ with $Fe_{sol}$% and EF of Pb could be explained by the influences from anthropogenic sources, according to l.348. Excluding the influence of nss-$SO_4^{2-}$ could mean that the anthropogenic factor is excluded. Thus, the residue could represent something other than the anthropogenic factor. Please justify the association of the residue with the emission from coal combustion. | As you point out, discussion about spurious correlations is likely suspicious. We have improved the sentences and removed the Fig. 5f (a scatter plot of residues). In the revised version, nss-$SO_4^{2-}$ was treated as a mediator variance rather than a conjunction factor (details are described in L371-377). Revised text as it appears in the text are described below: L.353-361: Moreover, L-Fe was extracted with MQ water in this study (weakly acidic to neutral conditions), but Fe in the coal fly ash is hardly soluble under these conditions ($Fe_{sol}$% < 0.2%, Desboefus et al., 2005; Oakes et al., 2012). Furthermore, all of the Fe in coal fly ash is not dissolved in acidic solutions ($Fe_{sol}$%: ~ 70% at pH 1.0, Chen and Grassian, 2013; Baldo et al., 2022). Therefore, if coal fly ash is the dominant L-Fe source, the EF of Co in the aerosol |

| | | | |
|---|---|---|---|
| | | should be higher than 3.0. However, the EFs of Co in the $PM_{1.3}$ samples were approximately 1.0 (Fig. S2). These results indicated that Fe in size-fractionated aerosol particles were mainly derived from mineral dust rather than coal fly ash and anthropogenic Fe oxides. However, $Fe_{sol}\%$ in non-aged mineral dust is usually less than 1.0% in weakly acidic and neutral solutions. Therefore, high $Fe_{sol}\%$ in $PM_{1.3}$ were caused by atmospheric processes of mineral dust during the transport.

L.372-378: It should be noted that the correlation between $Fe_{sol}\%$ and EF of Pb was caused by a high correlation between $nss\text{-}SO_4^{2-}$ and EF of Pb (Fig. 5e). Considering the causal relationship between $Fe_{sol}\%$ and EF of Pb, it is difficult to believe that $Fe_{sol}\%$ increases with increasing emissions of coal fly ash (increasing EF of Pb) because Fe in coal fly ash is insoluble unless the fly ash undergoes acidification (Desboefus et al., 2005; Oakes et al., 2012). Therefore, it seems that coal-derived $SO_2$ or $H_2SO_4$ emitted with Pb by coal combustion solubilizes Fe, resulting in the correlation between $Fe_{sol}\%$ and EF of Pb that may have occurred with $nss\text{-}SO_4^{2-}$ as a mediator variance. | | |
| 5. | l.376: EF of Co for S6-CPO is about 40 (Fig. S2). Thus, S6-CPO could be influenced by anthropogenic sources. No terrestrial influence from the 10-day backward trajectories cannot deny this. | One reason for the high EF of Co in CPO samples may be contamination during sample processing. We have established an ultra-clean aerosol sampling method, and carefully treat aerosol samples during experiment. Even so, the possibility of contamination cannot be completely excluded. In the case of CPO samples, even a small degree of contamination significantly impacts analytical data due to low metal concentrations. We think that the filter blanks and operating blanks in our method are among the best in the world, but still it is not easy to overcome this problem with the current technology.
Indeed, the absence of terrestrial influences cannot rule out anthropogenic influences. In contrast, the terrestrial influences do not guarantee the impact of anthropogenic emissions because the CPO is in the Southern Hemisphere, far from areas of high anthropogenic activity. At this stage, it is not easy to provide any clear answer. As in your comment in Round 1 (comment 2.2), if comparisons between our observations and a previous model study (Hamilton et al., 2019) are allowed, the contribution of anthropogenic FE to the CPO is small | | |

| | | (~5%, Hamilton et al., 2019). Therefore, the anthropogenic contribution cannot be emphasized in this region. |
|---|---|---|
| 6. | l.515: Please correct to under. | Thank you for pointing this out. We have corrected it. |